# Improved Regret Bounds in Stochastic Contextual Bandits with Graph Feedback

## Abstract

This paper investigates the stochastic contextual bandit problem with general function space and graph feedback. We propose a novel algorithm that effectively adapts to the time-varying graph structures, leading to improved regret bounds in stochastic settings compared with existing approaches. Notably, our method does not require prior knowledge of graph parameters or online regression oracles, making it highly practical and innovative. Furthermore, our algorithm can be modified to derive a gap-dependent upper bound on regrets, addressing a significant research gap in this field. Extensive numerical experiments validate our findings, showcasing the adaptability of our approach to graph feedback settings. The numerical results demonstrate that regrets of our method scale with graph parameters rather than action set sizes. This algorithmic advancement in stochastic contextual bandits with graph feedback shows practical implications in various domains.

## 1 Introduction

The bandit framework has garnered significant attention from the online learning community due to its widespread applicability in diverse fields such as recommendation systems, portfolio selection, and clinical trials (Li et al., 2010). Among the significant aspects of sequential decision making within this framework are side observations, which can be feedback from multiple sources (Mannor & Shamir, 2011) or contextual knowledge about the environment (Abbasi-Yadkori et al., 2011; Agarwal et al., 2012). These are typically represented as graph feedback and contextual bandits respectively.

The multi-armed bandits framework with feedback graphs has emerged as a mature approach, providing a solid theoretical foundation for incorporating additional feedback into the exploration strategy (Alon et al., 2017; Caron et al., 2012; Alon et al., 2015). The contextual bandit problem is another well-established framework for decision-making under uncertainty (Lattimore & Szepesvári, 2020; Chu et al., 2011; Abbasi-Yadkori et al., 2011). Despite the considerable attention given to non-contextual bandits with feedback graphs, the exploration of contextual bandits with feedback graphs has been limited (Zhang et al., 2023; Wang et al., 2021; Singh et al., 2020). Notably, Zhang et al. (2023) presented the first solution for general feedback graphs and function classes in the semi-adversarial settings, providing a minimax upper bound on regrets. However, to the best of our knowledge, there is no prior work considering stochastic feedback graphs and contexts while providing regret guarantees for general reward functions and feedback graphs.

In this paper, we fill this gap by proposing a practical graph learning algorithm that can simultaneously adapt to time-varying graph structures. We introduce a probabilistic sampling method that focuses on informative actions discovered through graph structures and empirical evaluation. Importantly, our method does not require prior knowledge of difficult-to-obtain graph parameters like the independence number. We also show the minimax optimality of our algorithm (up to logarithmic terms in the time horizon) by establishing a regret lower bound. To achieve this, we construct a series of bandit instances with a policy space that exhibits small variations across contexts. By controlling the number of observations using constraints based on the number of action selection and the underlying graph structures, we can prove the lower bound in the new setting.

The contributions of this paper are as follows:

1. We introduce a novel algorithm for contextual bandits with a general reward function space and probabilistic feedback graphs. This algorithm unifies the results from previous studies

(Wang et al., 2021; Singh et al., 2020) and improve regret bounds in stochastic settings compared to existing approaches (Zhang et al., 2023). Our algorithm exhibits adaptability to time-varying feedback graphs and contextual bandit instances, making it well-suited for stochastic settings. Additionally, the algorithm can be modified to derive a gap-dependent upper bound on regrets, addressing a significant research gap in (Zhang et al., 2023).

2. Besides the improved regret bounds, we further highlight the novelty of our algorithm through enhanced implementation practicability. Unlike existing methods, it does not require prior knowledge of graph parameters, simplifying the implementation process (Wang et al., 2021; Zhang et al., 2023). It also provides advantages in terms of offline regression oracles compared to methods relying on complex online regression oracles (Zhang et al., 2023). Furthermore, our proposed approach is computationally efficient compared to existing methods that involve solving nonlinear (Singh et al., 2020) and convex (Zhang et al., 2023) optimization problems. We also demonstrate that the convex optimization problem of Zhang et al. (2023) is numerically unstable in Appendix A.2 and Section 1.2, emphasizing the computational efficiency and stability of our algorithm.

3. Extensive numerical experiments are conducted to validate the findings. Our ablation experiments necessitate the special algorithmic design in graph feedback setting and showcase the adaptability of our approach by testing on randomly generated graphs. Further experiments demonstrate that the regret of the proposed method scales with the graph parameters rather than action set sizes.

| Works | Function Space | Required Prior Knowledge | Graph Type | Computational Cost |
|---|---|---|---|---|
| MAB (Buccapatnam et al., 2014) | no | informed graph | fixed, probabilistic | LP |
| linear contextual bandit (Singh et al., 2020) | linear | informed graph | fixed, deterministic | non-linear optimization |
| linear contextual bandit (Wang et al., 2021) | linear | informed graph, independence number at each round | time-varying, deterministic | not required |
| contextual bandit (Zhang et al., 2023) | general function space $\mathcal{F}$ | informed graph, uniform upper bound of independence numbers | time-varying, deterministic | convex optimization |
| contextual bandit (ours) | general function space $\mathcal{F}$ | informed graph | time-varying, probabilistic | greedy algorithm or LP |

Table 1: Comparison with existing works. Here, $\tilde{\mathcal{O}}$ hides the absolute constants and logarithmic terms in $T$. We provide comprehensive comparison in Section 1.2.

| Works | Gap-dependent Regret Upper Bound | Gap-independent Regret Upper Bound | Lower Bound |
|---|---|---|---|
| MAB (Buccapatnam et al., 2014) | $\tilde{\mathcal{O}}(\frac{\delta_f(G)}{\Delta})$ | not provided | $\Omega(\frac{\delta_f(G)\log T}{\Delta})$ (stochastic) |
| linear contextual bandit (Singh et al., 2020) | $\tilde{\mathcal{O}}(\frac{d\delta_f(G)}{\Delta})$ | not provided | $\Omega(\frac{d\delta_f(G)\log T}{\Delta})$ (stochastic) |
| linear contextual bandit (Wang et al., 2021) | not applicable | $\tilde{\mathcal{O}}(\sqrt{d\mathbb{E}_G[\alpha(G)]T})$ | not provided (adversarial) |
| contextual bandit (Zhang et al., 2023) | not applicable | $\tilde{\mathcal{O}}(\sqrt{\alpha T\log(|\mathcal{F}|)})$ | not provided (adversarial) |
| contextual bandit (ours) | $\tilde{\mathcal{O}}(\frac{\theta^{pol}\delta_f(G)\log(|\mathcal{F}|)}{\Delta})$ | $\tilde{\mathcal{O}}(\sqrt{\mathbb{E}_G[\delta_f(G)]T\log(\delta^{-1}|\mathcal{F}|)})$ | $\sqrt{\mathbb{E}_G[\delta_f(G)]T\log|\mathcal{F}|}$ (stochastic) |

Table 2: Continued table of table 1. Note $\alpha \geq \mathbb{E}_G[\alpha(G)]$.

## 1.1 RELATED WORK

Our work is closely related to contextual bandits, which have been extensively studied due to their wide range of applications, including scheduling, dynamic pricing, packet routing, online auctions, e-commerce, and matching markets (Cesa-Bianchi & Lugosi, 2006). Several formulations of contextual

bandits have been proposed, such as linear bandits (Abbasi-Yadkori et al., 2011), generalized linear bandits (Abbasi-Yadkori et al., 2011; Li et al., 2017; Chu et al., 2011), and kernelized bandits (Valko et al., 2013; Zenati et al., 2022). Researchers have designed algorithms tailored to specific function space structures, aiming to achieve near-optimal regret rates. A comprehensive summary of bandit algorithms can be found in the book by Lattimore & Szepesvári (2020). More recently, a line of works (Foster et al., 2018; Agarwal et al., 2012; Foster & Rakhlin, 2020; Gong & Zhang, 2023) has achieved optimal regret bounds for contextual bandits with a general function class, assuming access to a regression oracle. These findings are novel, requiring only minimal realizability assumptions. Our work builds upon these advancements and extends them to the setting of graph feedback.

Our work also relates to online learning with side information modeled by feedback graphs. The concept of feedback graphs was first introduced by Mannor & Shamir (2011) as a way to interpolate between full information settings, where the agent is aware of all rewards at each round, and bandit settings, where only the reward of the selected action is known. Online learning with feedback graphs has been extensively analyzed by Alon et al. (2015; 2017) and other authors (Rouyer et al., 2022; Chen et al., 2021; Cohen et al., 2016; Cesa-Bianchi et al., 2013; Atsidakou et al., 2022). Various methods, including UCB (Lykouris et al., 2020; Caron et al., 2012), TS (Lykouris et al., 2020), EXP (Rouyer et al., 2022; Chen et al., 2021; Alon et al., 2015; Cohen et al., 2016), IDS (Liu et al., 2018), and their variants, have been designed for this setting. Notably, Dann et al. (2020) extended the concept to reinforcement learning.

## 1.2 COMPARISON TO EXISTING WORKS

Our proposed approach offers significant advancements in the field of contextual bandits with graph feedback, particularly when compared to the prior work of Singh et al. (2020); Wang et al. (2021); Zhang et al. (2023). While Singh et al. (2020); Wang et al. (2021); Zhang et al. (2023) introduced valuable contributions, our method surpasses their results by providing substantially improved regret upper bounds. Furthermore, our algorithm demonstrates superior adaptability to both graph structures and bandit instances. In this section, we compare our approach to Singh et al. (2020); Wang et al. (2021); Zhang et al. (2023) in terms of regret bounds, algorithmic design, graph types, and instance-dependent versions of regrets. Through this comparison, we highlight the strengths of our method and address the limitations of existing approaches. For notation brevity, we use $\tilde{\mathcal{O}}$ to hide the polynomial of logarithmic terms in time horizon $T$.

**Comparison to (Singh et al., 2020).** Singh et al. (2020) investigate contextual bandits with side-observations and a linear payoff reward function. They propose a UCB-type learning algorithm, which provides gap-dependent upper bounds scaling with $\tilde{\mathcal{O}}(\frac{\Delta_{max}}{\Delta_{min}^2}\delta(G)d)$ (Corollary 1 in (Singh et al., 2020)), where $\Delta_{min}$ (equal to $\Delta$) and $\Delta_{max}$ represent the uniform lower and upper gaps across arms, respectively. Additionally, $d$ denotes the dimension of linear function spaces, and $\delta(G)$ is the dominating number of the graph $G$.

In our work, algorithm 1(algorithm 5) and Theorem 3.1(Theorem 3.3) can be easily extended to infinite function spaces $\mathcal{F}$ by standard learning-theoretic complexity measures such as metric entropy. We can consider a $\epsilon$-covering $\mathcal{F}_\epsilon$ of $\mathcal{F}$. Since $|\mathcal{F}_\epsilon|$ is finite, we can directly replace $\mathcal{F}$ with $\mathcal{F}_\epsilon$ without changing any algorithmic procedures, incurring an additional cost of $\epsilon T$ in regrets. The metric entropy of $d$-dimensional linear spaces is $\mathcal{O}(d\log(\frac{1}{\epsilon}))$ (Fan, 1953). By setting $\epsilon = \frac{1}{T}$, we obtain regret upper bounds of $\tilde{\mathcal{O}}(\frac{\delta_f(G)d}{\Delta})$(Theorem 3.1) and $\tilde{\mathcal{O}}(\sqrt{\delta_f(G)dT})$(Theorem 3.3). It is worth noting that for any contextual bandit instance and any graph $G$, we have the inequality: $\frac{1}{\Delta} \leq \frac{\Delta_{max}}{\Delta_{min}^2}, \delta_f(G) \leq \delta(G)$. Therefore, our gap-dependent bounds have a better dependence on instance and graph parameters.

Furthermore, it is important to highlight that (Singh et al., 2020) deals exclusively with fixed feedback graphs, while our work considers time-varying graphs which contains fixed graphs as a special case. As (Singh et al., 2020) focuses on a single graph, they can plan exploration steps and achieve a $\mathcal{O}(\delta(G))$ dependence on regrets. However, this requires a computationally intense step: solving a non-linear optimization problem to determine the fraction of exploration for each arm. Compared with their approach, we only need to solve an LP problem.

**Comparison to (Wang et al., 2021).** The paper (Wang et al., 2021) focuses on adversarial graphical contextual bandits, which incorporates two types of common side information: contexts and graph feedbacks. They propose EXP-based algorithms that leverage both contexts and graph feedbacks to

achieve improved performance. Similar to our previous arguments, we can derive minimax regret upper bounds in terms of linear function spaces.

In terms of regret bounds, both (Wang et al., 2021) and our algorithm 1 with the first option can incur $\tilde{\mathcal{O}}(\sqrt{\mathbb{E}_G[\alpha(G)]dT})$ regrets, where $\mathbb{E}_G[\alpha(G)]$ represents the expected independence number. However, a key distinction arises when considering time-varying graphs. Wang et al. (2021) requires knowledge of the independence number $\alpha(G_t)$ at each round to adapt to the changing graphs. They rely on these quantities to tune the input learning rate before the learning process starts. Actually, obtaining the independence number for a given graph is an NP problem (Busygin & Pasechnik, 2006), making this approach impractical. However, the minimax regrets in our work can be improved to $\tilde{\mathcal{O}}(\sqrt{\mathbb{E}_G[\delta_f(G)]dT})$, which is guaranteed to be better in graph parameters. Note that we still do not need to know the minimum dominating set or dominating number in algorithm 1 with the second option. Our required information is the pre-informed graph $G_t$, the same as (Wang et al., 2021).

**Comparison to (Zhang et al., 2023)**. The work of (Zhang et al., 2023) presents the first solution to contextual bandits with general function classes and graph feedback, resulting in the minimax regrets of $\tilde{\mathcal{O}}(\sqrt{\alpha T \log |\mathcal{F}|})$. Here, $\alpha$ represents the uniform upper bound for each feedback graph, i.e., $\alpha \geq \alpha(G_t)$. While (Zhang et al., 2023) provides important contributions, our approach with both options achieves substantially better regret upper bounds, due to $\alpha \geq \mathbb{E}_G[\alpha(G)]$ and $\alpha(G) \geq \delta_f(G)$ for any graphs. This improvement stems from our algorithm's ability to adapt to time-varying graphs.

The approach in (Zhang et al., 2023) tackles the problem by reducing it to an online regression oracle. However, designing and implementing online oracles are more challenging than offline oracles due to their sensitivity to the order of data sequence. In practice, the simple least square estimators in (Wang et al., 2021; Singh et al., 2020) can serve as valid offline oracles but not valid online oracles. Zhang et al. (2023) proposing solve the convex optimization problem to achieve near-optimal regrets:

$$\min_{p,z} p^\top \hat{\boldsymbol{\Delta}} + z$$
$$s.t. \ \forall a \in \mathcal{A} : \frac{1}{\gamma} \|p - e_a\|_{diag(G^\top p)^{-1}} \leq \hat{\Delta}_a + z, G^\top p \succ 0, \mathbf{1}^\top p = 1, \tag{1}$$

where $\hat{\boldsymbol{\Delta}}$ is a vector for $\{\hat{\Delta}_a, a \in \mathcal{A}\}$, $e_a$ is the $a$-th standard basis vector, $\succ$ means element-wise greater, and $\mathbf{1} \in \mathbb{R}^{|\mathcal{A}|}$ is a all-one vector. The above optimization (1) also incorporates a global hyperparameter $\gamma$, requiring prior knowledge of the uniform upper bound $\alpha$ for suitable tuning.

However, it has been observed that the Python code provided by Zhang et al. (2023) for solving optimization problems suffers from numerical instability. This instability manifests in the form of inaccurate solutions and the inability to generate feasible solutions for certain instances, as demonstrated in Appendix A.2. The root cause of this numerical instability can be traced back to the first constraint in their formulation, which involves the inversion of the unknown variable $p$. Particularly, when certain components of the optimal $p$ become very small, the computation of the inverse of $p$ becomes highly inaccurate and numerically unstable.

Zhang et al. (2023) provide some tricks to mitigate the numerical instability for large values of $\gamma$ (i.e., large $T$). Readers might think that powerful Python packages for convex optimization could address this issue. However, we emphasize that the barrier could be fundamental. As the learning process progresses, the sampling probabilities for suboptimal arms gradually converge to zero. Therefore, it is inevitable that some components of the optimal $p$ will become extremely small, leading to the failure of the algorithm in (Zhang et al., 2023). In contrast, our algorithm overcomes this barrier by establishing an explicit sampling distribution on the selected exploartion set.

Finally, we emphasize that Zhang et al. (2023), as well as Wang et al. (2021), fail to generalize to gap-dependent upper bounds and probabilistic feedback graphs, which are significant contributions of our work. Adapting the sampling probability to specific bandit instances and time-varying graphs would require non-trivial modifications to the algorithmic design and proofs.

## 2 PROBLEM FORMULATION

Throughout this paper, we let $[n]$ denote the set $\{1, 2, \cdots, n\}$ for any positive integer $n$. We consider the following contextual bandits problem with informed feedback graphs. The learning

process goes in $T$ rounds. At each round $t \in [T]$, the environment independently selects a context $x_t \in \mathcal{X}$ and an undirected feedback graph $G_t \in \mathcal{G}$, where $\mathcal{X}$ is the domain of contexts and $\mathcal{G}$ is the collections of deterministic and undirected graphs with self-loops. Both $G_t$ and $x_t$ are revealed to the learner at the beginning of each round $t$. Let $\mathcal{A}$ be the action set, which consists of nodes of graphs in $\mathcal{G}$. For each $a \in \mathcal{A}$, denote the neighborhood of a node (arm) $a$ in $\mathcal{A}$ as $\mathcal{N}_a(G) = \{v \in \mathcal{A} : (v, a) \in E \text{ for } G = (|\mathcal{A}|, E)\}$. Then the learner selects one of the actions $a_t \in \mathcal{A}$ and then observes rewards according to the feedback graph $G_t$. Specifically, for the action $a_t$, the learner observes the reward of all actions in $\mathcal{N}_{a_t}(G_t)$, i.e., $\{(x_t, a, y_{t,a})\}$ for $a \in \mathcal{N}_{a_t}(G_t)$. We assume all rewards are bounded in $[0, 1]$ and all reward distributions are independent.

We assume that the learner has access to a class of reward functions $\mathcal{F} \subset \mathcal{X} \times \mathcal{A} \to [0, 1]$ (e.g., linear function classes) that characterizes the mean of the reward distribution for a given context-action pair. In particular, we make the following standard realizability assumption (Simchi-Levi & Xu, 2021; Foster & Rakhlin, 2020; Foster et al., 2020).

**Assumption 2.1 (realizability)** *There exists a function $f^* \in \mathcal{F}$ such that $f^*(x_t, a) = E[y_{t,a}|x_t]$ for all $a \in \mathcal{A}$ and all $t \in [T]$.*

The set of all induced policies is denoted as $\Pi = \{\pi_f | f \in \mathcal{F}\}$, which forms the policy space. The objective of the agent is to achieve low regret with respect to the optimal policy $\pi_{f^*}$, and the regret over time horizon $T$ is defined as follows: $Reg(T) = \sum_{t=1}^{T} \mathbb{E}[f^*(x_t, \pi_{f^*}(x_t)) - f^*(x_t, \pi_f(x_t))|x_t]$.

We further assume access to an offline least square regression oracle for function class $\mathcal{F}$. Based on the dataset $\{(x_n, a, y_{n,a}) | a \in \mathcal{N}_{a_n}(G_n)\}_{n=1}^{t-1}$, the goal of the oracle is to find an estimator $\hat{f}_t \in \mathcal{F}$ via minimizing the cumulative square errors: $\hat{f}_t = \arg\min_{f \in \mathcal{F}} \sum_{n=1}^{t-1} \sum_{a \in \mathcal{N}_{a_n}(G_n)} (f(x_n, a) - y_{n,a})^2$. Our paper only requires the offline regression oracle, compared with (Zhang et al., 2023) which requires online oracle. Though Foster & Krishnamurthy (2021) provide several examples of online regression algorithms, it is more efficient and simpler to design an algorithm under offline oracle.

## 3 ALGORITHM DESIGN AND REGRET ANALYSIS

---

**Algorithm 1** A FAster COntextuaL bandit algorithm with Graph feedback (FALCON.G)

---

**Input:** time horizon $T$, confidence parameter $\delta$, tuning parameters $\eta$
1: Set epoch schedule $\{\tau_m = 2^m, \forall m \in \mathbb{N}\}$
2: **for** epoch $m = 1, 2, \cdots, \lceil \log_2 T \rceil$ **do**
3:      Compute the function $\hat{f}_m = \arg\min_{f \in \mathcal{F}} \sum_{n=1}^{\tau_{m-1}} \sum_{a \in \mathcal{N}_{a_n}(G_n)} (f(x_n, a) - y_{n,a})^2$ via the **Offline Least Square Oracle**
4:      **for** round $t = \tau_{m-1} + 1, \cdots, \tau_m$ **do**
5:          Observe the context $x_t$ and the graph $G_t$
6:          Compute the best arm candidate set $\mathcal{A}(x_t)$ and find $\hat{a}_t = \max_{a \in \mathcal{A}} \hat{f}_m(x_t, a)$
7:          **if** $|\mathcal{A}(x_t)| == 1$ **then**
8:              Let the exploration set be $S_t = \{\hat{a}_t\}$
9:          **else**
10:              Call the subroutine **ConstructExplorationSet** to find the exploration set $S_t$
11:              **if** $|S_t| \geq |\mathcal{A}(x_t)|$ **then**
12:                  Let the exploration set $S_t$ be $\mathcal{A}(x_t)$
13:          Compute $\gamma_t = \sqrt{\frac{\eta |S_t| \tau_{m-1}}{\log(\delta^{-1}|\mathcal{F}|m\log(|\mathcal{A}|T))}}$ (for the first epoch, $\gamma_t = 1$)
14:          Compute the following probabilities

$$p_t(a) = \begin{cases} \frac{1}{|S_t| + \gamma_t(\hat{f}_m(x_t, \hat{a}_t) - \hat{f}_m(x_t, a))}, & \text{for all } a \in S_t - \{\hat{a}_t\} \\ 0, \text{ for all } a \in \mathcal{A} - S_t \\ 1 - \sum_{a \neq \hat{a}_t} p_t(a), \text{ for } a = \hat{a}_t \end{cases}$$

15:          Sample $a_t \sim p_t(\cdot)$ and take the action $a_t$
16:          Observe a feedback graph $\{(a, y_{t,a}) | a \in \mathcal{N}_{a_t}(G_t)\}$ from $G_t$

---

We present our method in algorithm 1, where the side-observations, including contextual information and graph feedbacks, are used nearly optimal. In more detail, we operate in a doubling epoch schedule. Letting $\tau_m = 2^m$ with $\tau_0 = 0$, each epoch $m \geq 1$ consists of rounds $\tau_{m-1} + 1, ..., \tau_m$, and there are $\lceil \log_2 T \rceil$ epochs in total. To understand the motivation behind the our designed algorithm, we point out several features of our algorithms.

**Adaptive to action sets.** At the beginning of each epoch $m$, algorithm 1 identifies the best empirical estimator $\hat{f}_m$ based on square loss using historical data. We define the best arm candidate set $\mathcal{A}(x) = \{a | a = \arg\max_{a \in \mathcal{A}} f(x, a) \text{ for some } f \text{ in } \mathcal{F}\}$. A systematic method of computing the candidate action set $\mathcal{A}(x)$ is provided in Section 4 and Appendix A of (Foster et al., 2020). Although one can replace $\mathcal{A}(x)$ with the entire action set $\mathcal{A}$ without changing the algorithmic procedures in algorithm 1, it can be beneficial to compute $\mathcal{A}(x)$ for improving numerical performance. However, it is crucial to compute $\mathcal{A}(x)$ in order to obtain gap-dependent upper bounds, as shown in algorithm 5.

As a result, our main objective is to ensure that each arm within $\mathcal{A}(x)$ is observed sufficiently often. The additional graph feedback enables us to gather information on certain arms by playing adjacent arms in the graph. Leveraging this property of feedback graphs, we restrict exploration to a subset of arms while still obtaining sufficient information about all the arms in $\mathcal{A}(x)$. The exploration set $S_t$ is constructed by the subroutine **ConstructExplorationSet**, which is used by algorithm 1 to compute the sampling probabilities on each arm. There are two options for finding a feasible subset, each leading to different theoretical guarantees regarding graph parameters (See Appendix C).

• **Option 1**: **ConstructExplorationSet**($G_t$, $\mathcal{A}(x_t)$, $x_t$, $\hat{f}_m$). The first option of the subroutine **ConstructExplorationSet** aims to find an independence set within the induced subgraph of $G_t$ for the arms in $\mathcal{A}(x_t)$. The subroutine begins by sorting the arms in ascending order based on their gap estimates. Subsequently, $S_t$ is constructed greedily by iteratively selecting the arm with the smallest gap and removing its neighbors in $G_t$ from consideration. Since the chosen arms are non-adjacent in $G_t$, this procedure yields an independence set. The size of $S_t$ is thus bounded by the independence number of $G_t$. Intuitively, by prioritizing arms with smaller estimated gaps, the subroutine efficiently explores the most promising regions of the action space while respecting the dependence structure in $G_t$, as imposed by its independence set construction.

• **Option 2**: **ConstructExplorationSet**($G_t$, $\mathcal{A}(x_t)$). The second option for implementing the **ConstructExplorationSet** involves solving a linear optimization problem. To find the most efficient exploration set, we formulate the following linear problem:

$$\begin{aligned} & \mathbf{1}^\top \mathbf{z} \\ \text{s.t.} \quad & G_t \mathbf{z} \geq \mathbf{1} \\ & \mathbf{z} \geq \mathbf{0}. \end{aligned} \tag{2}$$

Here, we denote $G_t$ as the adjacency matrix of the graph, and the optimal value of this linear program corresponds to the fractional dominating number $\delta_f(G_t)$. However, one challenge is that the optimal solution may involve a "fractional number of pulls" for arms. To address this challenge, we make use of Bernoulli trials and exploit the following observation: the components of the optimal solution to (2) are within the range of $[0, 1]$. Hence, the expected size of the exploration set $S_t$ is $\delta_f(G_t)$. To further refine the exploration set $S_t$ and reduce the number of arms, we perform the following adjustments. First, any arms in $S_t$ whose neighbors do not cover any arms in $\mathcal{A}(x_t)$ are removed. Additionally, for each arm in $S_t$, if its neighbor contains only one arm from $\mathcal{A}(x_t)$, it is replaced with the only adjacent arm in $\mathcal{A}(x_t)$. Finally, if the empirical best arm has not been selected yet, it is added to $S_t$. This process ensures the construction of an exploration set $S_t$ with its expected cardinality of at most $\delta_f(G_t) + 1$.

Our algorithm is based on the inverse gap weighting technique (IGW) (Foster et al., 2018; Agarwal et al., 2012; Foster et al., 2020; Simchi-Levi & Xu, 2021). However, our algorithm distinguishes itself from existing literature in two crucial aspects. Firstly, we only apply the IGW scheme to arms in the exploration set $S_t$, leveraging the graph feedback structure. This allows us to disregard the arms in $\mathcal{A} - S_t$ because they do not require exploration or can be explored freely. Secondly, we select the learning rate $\gamma_t$ to be adaptive to the exploration set $S_t$. This approach enables our algorithm to adapt to the given graph $G_t$ without prior knowledge of specific graph quantities. Unlike approaches such as (Zhang et al., 2023) that rely on pre-determined graph parameters like the uniform independence bound $\alpha$ or the independence number $\alpha(G_t)$ at each round (Wang et al., 2021), our

algorithm dynamically adapts to a dominating set (or fractional dominating set) of the graph $G_t$. As a result, the expected regret of our algorithm in the first option can scale with the expected independence number, rather than the uniform upper bound of independence numbers (Zhang et al., 2023). Moreover, with suitable modifications, our approach can achieve improved scaling with the expected fractional dominating number in the second option.

**Trade-off between gaps and arms.** The greedy construction of exploration set in the first option show the adaptive feature in gaps. By putting arms with small gaps in priority, we may improve the numerical performance of algorithm 1. The first option explores arms with the empirically minimal regrets, resulting in potentially low regrets despite exploring numerous arms. Yet, the second option chooses to explore the least times of arms compared with the first option, at a risk of suffering from high regrets for explorations. Actually, the highly informative arms in the exploration set may yield low returns, leading to a trade-off between the number of arms to explore and the reward gaps associated with those arms. As we mentioned in the second option, we refine the exploration set by replacing the obtained sets with the empirically better arms if the number of the latter is fewer. The following Lemma 3.1 shows that this operation does not change the (expected) size of $|S_t|$.

**Lemma 3.1** *For each epoch $m$ and the round $t$ in this epoch, the size of $S_t$ constructed in **Option 1** is bounded by $\alpha(G_t)$; the expected size of $S_t$ constructed in **Option 2** is bounded by $\delta_f(G_t) + 1$.*

To show the optimality of our algorithms, we prove the following minimax regret upper bounds.

**Theorem 3.1** *Consider a contextual bandit problem with graph feedbacks. Suppose that the realizability Assumption 2.1 holds. Then with probability at least $1 - \delta$, the expected regret $\mathbb{E}[Reg(T)]$ of algorithm 1 is upper bounded by (**Option 1**) $\mathcal{O}\left(\sqrt{\mathbb{E}_G[\alpha(G)]T\log(\delta^{-1}|\mathcal{F}|\log T\log(|\mathcal{A}|T))}\right)$; (**Option 2**) $\mathcal{O}\left(\sqrt{\mathbb{E}_G[\delta_f(G)]T\log(\delta^{-1}|\mathcal{F}|\log T\log(|\mathcal{A}|T))}\right)$. Here, the expectation of regrets is taken with respect to all randomness, i.e., the randomness of the environment and algorithm.*

Algorithm 1 with the first option is optimal for a large class of graphs, such as claw-free graphs (Bollobás & Cockayne, 1979). In these graphs, $\alpha(G)$ can be upper bounded by $(1 + \log|\mathcal{A}|)\delta_f(G)$ (Domke et al., 2017). Therefore, the first option incurs only an $\mathcal{O}(\log|\mathcal{A}|)$ factor in terms of regrets.

**Extensions: probabilistic feedback graphs and gap-dependent upper bounds.**

*Probabilistic feedback graphs.* We begin by showcasing the applicability of our algorithm, specifically the second option, to probabilistic feedback graphs. In this setting, when the learner selects an action $a$, it can observe the rewards of arms in $\mathcal{N}_a(G)$ with certain probabilities. These probabilities are determined by the adjacency matrix $G := (g_{ij})$, where each entry $g_{ij}$ represents the probability of observing the reward of action $j$ when the action $i$ is chosen. It is important to note that deterministic graphs are special cases of probabilistic graphs, where the probabilities take binary values (0 or 1). We still focus on self-aware (i.e., $g_{ii} = 1$ for each action $i$) and undirected (i.e., $g_{ij} = g_{ji}$) graph collection $\mathcal{G}$. Furthermore, we consider the informed and stochastic settings, which are prevalent in graph feedback scenarios (Cortes et al., 2020; Ghari & Shen, 2023; Kong et al., 2022; Li et al., 2020).

To address the challenges posed by general feedback graphs where the independence set or dominating set is not well-defined, we focus on the second option. We denote $G_t$ as the probabilistic adjacency matrix of the feedback graph $G_t$, and the expression of (2) remains unchanged. We still refer to the optimal value of (2) as the probabilistic fractional dominating number, denoted as $\delta_f(G_t)$. To tackle this challenge, we utilize geometric random variables and extend the exploration set $S_t$ to an exploration multiset, allowing for multiple elements in $S_t$. The theorem regarding the minimax upper bound of algorithm 1 still applies to the second option. We then demonstrate that this upper bound matches the lower bound in the general setting.

Several previous studies, such as (Alon et al., 2017) and (Alon et al., 2015), have established an information-theoretic lower bound on the minimax regret for adversarial graph bandit problems. The fundamental idea behind these bounds is to select a maximal independence set and transform the original $|\mathcal{A}|$-armed bandit learning problem into $\alpha$-armed bandit problems. By exploiting the properties of the independence set, the rewards of arms outside the selected set are effectively set to zero. Consequently, no algorithm can achieve a performance better than $\sqrt{\alpha T}$ in the adversarial settings. In the stochastic setting, we can plan our exploration strategy more efficiently as there does

not exist adversary. By leveraging the idea, we can also prove the following lower bound in stochastic graph bandit setting.

**Theorem 3.2** *There exists an instance with $|\mathcal{A}|$ actions, the function class $\mathcal{F}$ and the time-invariant feedback graph $G$ such that the expected regret is lower bounded by $\Omega(\sqrt{\delta_f(G)T\log|\mathcal{F}|})$. As a corollary, there exists an instance with time-varying graphs $G_t$ such that the expected regret is lower bounded by $\Omega(\sqrt{\mathbb{E}_G[\delta_f(G)]T\log|\mathcal{F}|})$.*

Our algorithm 1 can be applicable to multi-armed bandit (MAB) instances, and it retains the validity of Theorem 3.3. Notably, in comparison to the MAB algorithm proposed in (Buccapatnam et al., 2017), our approach achieves regrets of order $\mathcal{O}\left(\sqrt{\mathbb{E}_G[\delta_f(G)]T\log(\delta^{-1}|\mathcal{F}|\log^2 T)}\right)$, with an additional factor $\log|\mathcal{F}|$. The algorithm proposed in (Buccapatnam et al., 2017) typically involves two phases: an exploration phase that maximizes the utilization of the graph structure, and an exploitation phase where arms are selected based on the principle of optimism in the face of uncertainty. However, the reliance of their method on fixed graph structures during the exploration phase limits its applicability to scenarios involving time-varying graphs. In contrast, our algorithm 1 is capable of handling time-varying graphs with the cost of additional factor in the expected regret.

It is important to note that our derived lower bound does not directly apply to MAB settings, because the realizability assumption is not required to hold in the MAB framework. This condition assumes the MAB learner can find the true reward vector within the range of $[0,1]^{|\mathcal{A}|}$, while the primary objective in MAB settings is to identify the arm with the highest mean reward. Therefore, even if $\mathcal{X}$ is a singleton, the realizability assumption extends beyond identifying the arm with the highest mean reward, thereby intensifying the complexity of our problem compared to conventional MAB settings. As a result, it is expected that our lower bound scales with an additional $\log|\mathcal{F}|$ term in comparison to pure graphical MAB settings (Buccapatnam et al., 2014; 2017).

*Gap-dependent upper bound.* We then show that our algorithm can be modified to derive gap-dependent upper bound. To establish this, we start by making a standard uniform gap assumption in stochastic settings.

**Assumption 3.1 (Uniform Gap Assumption)** *For all $x \in \mathcal{X}$, there exists $\Delta > 0$ such that $f^*(x, \pi_{f^*}(x)) - f^*(x, a) \geq \Delta, \forall a \neq \pi_{f^*}(x)$.*

However, it has been pointed out by (Foster et al., 2020) that obtaining a gap-dependent regret for general contextual bandits is not possible, even under the uniform gap assumption. To overcome this limitation, researchers have developed algorithms that achieve instance-dependent bounds for specific classes of problems under additional structural or distributional assumptions. Similar to (Foster et al., 2020), we consider the following policy disagreement coefficient $\theta^{pol}(\mathcal{F}, \epsilon_0) = \sup_{\epsilon \geq \epsilon_0} \frac{1}{\epsilon}\mathbb{P}_\mathcal{X}(x \in \mathcal{X} : \exists f \in \mathcal{F}_\epsilon$ such that $\pi_f(x) \neq \pi_{f^*}(x))$, where $\mathcal{F}_\epsilon = \{f \in \mathcal{F}|\mathbb{P}_\mathcal{X}(x \in \mathcal{X} : \pi_f(x) \neq \pi_{f^*}(x)) \leq \epsilon\}$. Foster et al. (2020) shows the policy disagreement coefficient is a sufficient and weak necessary condition under which the bandit algorithms can achieve the near-optimal regrets.

We modify the algorithm 1 to algorithm 5 which is presented in Appendix C. At the beginning of each epoch $m$, algorithm 5 builds an upper confidence bound for each function and maintain a function space $\mathcal{F}_m$, which is the set of all plausible predictors that cannot yet be eliminated based on square loss confidence bounds. Using the obtained $\mathcal{F}_m$, we compute the data-driven candidate action set as $\mathcal{A}(x; \mathcal{F}_m) = \{a|a = \arg\max_{a \in \mathcal{A}} f(x, a)$ for some $f$ in $\mathcal{F}_m\}$. To incorporate the gap dependence, we need to introduce the instance-dependent scale factor $\lambda_m$ to the parameter $\gamma_t$. This scale factor provides a sample-based approximation to the quantity: $\mathbb{P}_\mathcal{X}(|\mathcal{A}(x; \mathcal{F}_m)| > 1)/\sqrt{\mathbb{P}_\mathcal{X}(|\mathcal{A}(x; \mathcal{F}_{m-1})| > 1)}$. It can be observed that $\mathbb{P}_\mathcal{X}(|\mathcal{A}(x; \mathcal{F}_m)| > 1) \leq \mathbb{P}_\mathcal{X}(x \in \mathcal{X} : \exists f \in \mathcal{F}_m$ such that $\pi_f(x) \neq \pi_{f^*}(x))$, so the disagreement coefficient can provide a valid coefficient in upper bounds of the instance-dependent regrets.

**Theorem 3.3** *Consider a contextual bandit problem with graph feedbacks. Suppose that the realizability Assumption 2.1 holds. Then for any instance with uniform gap $\Delta > 0$, with probability at least $1 - \delta$, the expected regret $\mathbb{E}[Reg(T)]$ of algorithm 1 is upper bounded by (**Option 1**) $\mathcal{O}\left(\max\left\{\epsilon\Delta T, \frac{\theta^{pol}(\mathcal{F}, \epsilon)\mathbb{E}_G[\alpha(G)]\log(\delta^{-1}T^2|\mathcal{F}|)\log T}{\Delta}\right\}\right)$; (**Option 2**) $\mathcal{O}\left(\max\left\{\epsilon\Delta T, \frac{\theta^{pol}(\mathcal{F}, \epsilon)\mathbb{E}_G[\delta_f(G)]\log(\delta^{-1}T^2|\mathcal{F}|)\log T}{\Delta}\right\}\right)$.*

## 4 NUMERICAL EXPERIMENTS

**Ablation experiments.** The detailed numerical setup can be found in Appendix A. In this part, we use empirical results to demonstrate the significant benefits of algorithm 1 in leveraging the graph feedback structure. To demonstrate its effectiveness, we conduct numerical ablation experiments in time-varying graphs with both options. Our baseline is FALCON (Simchi-Levi & Xu, 2021), with additional observations only used for estimating $\hat{f}_m$ at the epoch $m$.

In our ablation experiments, we utilize algorithm 4 to generate time-varying random graphs with fixed dense factors. The results depicted in Figure 1 clearly demonstrate that the average regrets of algorithm 1 are significantly smaller than that of (Simchi-Levi & Xu, 2021). Intuitively, as random graphs become denser, their connectivity improves, leading to smaller independence numbers and fractional dominating numbers. However, the performance of (Simchi-Levi & Xu, 2021) does not improve as the random graphs become denser. In contrast, our algorithm 1 outperforms it in both options and exhibits improvement as the graphs become denser. These results highlight the adaptive capability of our approach to random graphs. In conclusion, our findings demonstrate that relying solely on additional observations to estimate $\hat{f}$ is insufficient. It is crucial to propose a specialized design for action sampling probabilities in graph feedback settings, as exemplified by our algorithm 1.

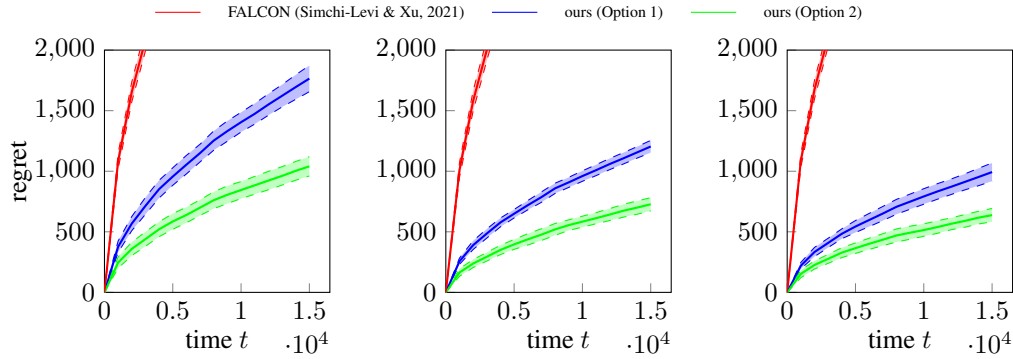

Fig. 1: Comparison with classical and our algorithms in graph feedback settings. Left: random graph(dense factor = 0.25); middle: random graph(dense factor = 0.5); right: random graph(dense factor = 0.75).

| size of the action set $\|\mathcal{A}\|$ | | 20 | 40 | 60 | 80 | 100 |
|---|---|---|---|---|---|---|
| mean(std) regret | option 1 | 1515(58) | 1445(58) | 1438(62) | 1396(43) | 1376(39) |
| at the final round $T$ | option 2 | 1742(93) | 1808(97) | 1799(103) | 1830(105) | 1810(87) |

Table 3: Test on clique-group with its clique number equal to 5.

**Test on graph parameters.** We test our algorithm in clique-group (as shown in Figure 2), where both options of algorithm 1 are optimal regarding graph quantities in our artificially constructed graphs. To showcase that our method does not scale with the size of the action set size, we report the average regret and its standard deviation at the final round in table 3.

Analyzing the results in table 3, we observe that the final regrets of both options fluctuate within one standard deviation. This indicates that the regrets do not scale with $\|\mathcal{A}\|$ since the clique number (equal to $\delta_f(G) = \alpha(G)$) remains fixed for each learning process. This numerical result validates the findings of our Theorem 3.1. Furthermore, we notice that the first option outperforms the second option, which aligns with our explanations in Section 3. In these types of graphs, the second option does not provide an advantage in reducing the number of explorations compared to the first option, while the first option can further adapt to instance gaps, resulting in reduced exploration regrets. Additionally, the final regret of the first option slightly decreases as the number of arms increases. This can be attributed to the fact that when the learner explores suboptimal arms, the rewards of these suboptimal arms are more likely to be high when the number of arms is larger. Hence, this serves as additional evidence of the capability of our algorithm with the first option to adapt to instance gaps. In contrast, the second option solely focuses on graph structures, and in table 3, there is no clear trend in the final regret as $\|\mathcal{A}\|$ increases.

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
