## A    OMITTED RESULTS

### A.1    EXPERIMENTAL SETUP

In numerical experiments, we randomly generate a function space $\mathcal{F} = \{(x - x_0)^\top (a - a_0)\}$ with a size of 50 by sampling $x_0$ and $a_0$ in $\mathbb{R}^d$ from standard normal distributions, where $d = 10$. We then randomly choose a function as the true reward function $f^*$ from $\mathcal{F}$, and generate the reward as

$$Y = f^*(X, A) + \mathcal{N}(0, 1),$$

where the context $X$ is drawn i.d.d. from $\mathcal{N}(0, 1)$ and $A$ is the selected action. The whole action set $\mathcal{A}$ is randomly initialized from $[-1, 1]^d$ with a size of 50. We repeat each instance 50 times to obtain a smooth regret curve. The hyperparameters in algorithm 1 are set to be $\eta = 1$ and $\delta = 0.1$

**Test on graph quantities.**    To demonstrate the effectiveness of our algorithms, we introduce a special type of graphs called clique-group (as shown in Figure 2). These graphs are designed such that each block consists of a perfect graph, resulting in $\delta_f(G) = \delta(G) = \alpha(G)$, as proven by Domke et al. (2017). Therefore, both options of algorithm 1 are optimal regarding graph quantities in our artificially constructed graphs. To showcase that our method does not scale with the size of the action set $|\mathcal{A}|$, we start with an instance where the size of $\mathcal{A}$ is 100. We then iteratively reduce the size of the action set to $80, 60, 40, 20$. We report the average regret and its standard deviation at the final round $T = 2^{14}$ in table 3.

In Section 4, we implement algorithms in various graphs to verify our findings. These graphs are created using our random graph generator, which is described in algorithm 4. To simplify the following graphs, self-loops are omitted. We present these graphs in Figure 2.

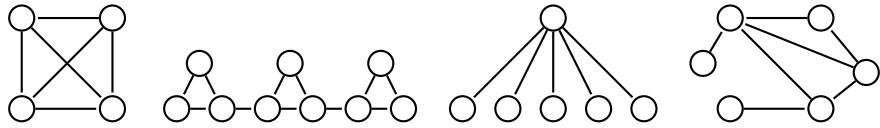

Fig. 2: Different types of graphs in order: a fully connected graph, a clique group, a k-tree, a random graph.

We also conducted numerical experiments on k-trees, where the graph has $\delta_f(G) = 1$ and $\alpha(G) = |\mathcal{A}| - 1$. In Figure 3, we observe that algorithm 1 with the first option has comparable performance with the baseline FALCON. The scaled graphical quantities for algorithm 1 and FALCON are $\sqrt{|\mathcal{A}| - 1}$ and $\sqrt{|\mathcal{A}|}$, respectively. However, algorithm 1 with the first option exhibits less variance as it occasionally reduces the size of the exploration set to 2. On the other hand, algorithm 1 with the second option shows a significant gap compared to the first option, highlighting its smaller leading constants in terms of regret.

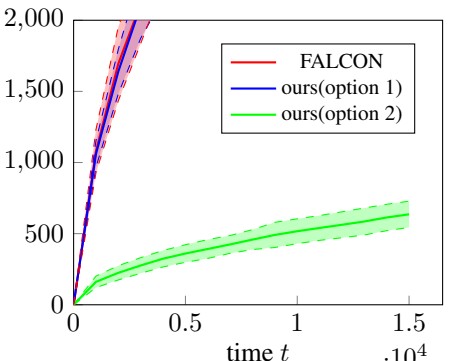

Fig. 3: Test on k-trees.

```
1   def makeProblem(nactions):
2       import cvxpy as cp
3       sqrtgammaG = cp.Parameter((nactions, nactions), nonneg=True)
4       sqrtgammafhat = cp.Parameter(nactions)
5       p = cp.Variable(nactions, nonneg=True)
6       sqrtgammaz = cp.Variable()
7       objective = cp.Minimize(sqrtgammafhat @ p + sqrtgammaz)
8       constraints = [
9           cp.sum(p) == 1
10      ] + [
11          cp.sum([ cp.quad_over_lin(eai - pi, vi)
12                  for i, (pi, vi) in enumerate(zip(p, v))
13                  for eai in (1 if i == a else 0,)
14              ]) <= sqrtgammafhata + sqrtgammaz
15          for v in (sqrtgammaG @ p,)
16          for a, sqrtgammafhata in enumerate(sqrtgammafhat)
17      ]
18      problem = cp.Problem(objective, constraints)
19      assert problem.is_dcp(dpp=True) # proof of convexity
20      return problem, sqrtgammaG, sqrtgammafhat, p, sqrtgammaz
```

Fig. 4: Python code of solving convex optimization problem in Zhang et al. (2023)

```
1   K = 10
2   Gr = [[1., 1., 0., 0., 0., 0., 0., 0., 0., 0.],
3    [1., 1., 1., 0., 0., 0., 0., 0., 0., 0.],
4    [0., 1., 1., 1., 0., 0., 0., 0., 0., 0.],
5    [0., 0., 1., 1., 1., 0., 0., 0., 0., 0.],
6    [0., 0., 0., 1., 1., 1., 0., 0., 0., 0.],
7    [0., 0., 0., 0., 1., 1., 1., 0., 0., 0.],
8    [0., 0., 0., 0., 0., 1., 1., 1., 1., 1.],
9    [0., 0., 0., 0., 0., 0., 1., 1., 1., 1.],
10   [0., 0., 0., 0., 0., 0., 1., 1., 1., 1.],
11   [0., 0., 0., 0., 0., 1., 1., 1., 1., 1.]]
12  estimator_reward = [3.85542284, -24.56530271,  -3.33500128,  -7.28226539,
13                      -6.7804921, -0.53989755,  -3.07904049,   0.35991018, -13.23318228, 196.46494904]
14  gamma = 3.5950994182855642
15
16  Gr = np.array([np.array(row) for row in Gr])
17  estimator_reward = np.array(estimator_reward)
18
19
20  problem, sqrtgammaG, sqrtgammafhat, p, sqrtgammaz = makeProblem(K)
21  sqrtgammaG.value = Gr * np.sqrt(gamma)
22  sqrtgammafhat.value = (max(estimator_reward) - estimator_reward) * np.sqrt(gamma)
23  problem.solve(verbose=True)
```

Fig. 5: An example that cannot be successfully solved by the Python code in Figure 4

## A.2 EXAMPLE OF NUMERICAL INSTABILITY

We use the Python code provided by Zhang et al. (2023) to solve the proposed convex optimization problem. The full version is in Figure 4. In Figure 5, we provide an example that cannot be successfully solved by the Python code in Figure 4.

## B PROOFS

### B.1 NOTATIONS

Our regret analysis builds on a framework established in Simchi-Levi & Xu (2021); Foster et al. (2020), which analyzes contextual bandit algorithms in the universal policy space $\Psi := \prod_{x \in \mathcal{X}} \mathcal{A}$. In our work, we consider a subspace of $\Psi$ reduced by graph feedbacks at each epoch and prove the sampling probability $p_t$ satisfy good properties to attain low regrets. Specifically, we consider the subspace $\Psi(S)$ where the policy is supported on $S$. With abuse of notations, we define

$$\mathcal{R}(\pi) = \mathbb{E}_x[f^*(x, \pi(x))] \text{ and } Reg(\pi) = \mathcal{R}(\pi_{f^*}) - \mathcal{R}(\pi).$$

The above quantities do not depend on specific values of $x$. The following empirical version of above quantities are defined as

$$\widehat{\mathcal{R}}_t(\pi) = \mathbb{E}_{x_t}[\hat{f}_{m(t)}(x_t, \pi(x_t))] \text{ and } \widehat{Reg}_t(\pi) = \mathbb{E}_x[\widehat{\mathcal{R}}_t(\pi_{\hat{f}_{m(t)}}) - \widehat{\mathcal{R}}_t(\pi)],$$

where $m(t)$ is the epoch of the round $t$.

For any realization $S_t$, $\gamma_t$ and $\hat{f}_m$, let $Q_t(\cdot)$ be the equivalent policy distribution for $p_t(\cdot|\cdot)$, i.e.,

$$Q_t(\pi) = \prod_{x \in \mathcal{X}} p_t(\pi(x)|x, S_t), \forall \pi \in \Psi(S_t).$$

The existence and uniqueness of such measure $Q_t(\cdot)$ is a corollary of Kolmogorov's extension theorem. Note that both $\Psi$ and $Q_t(\cdot)$ are $\mathcal{H}_{t-1}$-measurable, where $\mathcal{H}_{t-1}$ is the filtration up to the time $t - 1$. We refer to Section 3.2 of Simchi-Levi & Xu (2021) for more detailed intuition for $Q_t(\cdot)$ and proof of existence. By Lemma 4 of Simchi-Levi & Xu (2021), we know that for all epoch $m$ and all rounds $t$ in epoch $m$, we can rewrite the expected regret in terms of our notations as

$$\mathbb{E}[Reg(T)] = \sum_{t=1}^{T} \sum_{\pi \in \Psi} Q_t(\pi) Reg(\pi).$$

For simplicity, we define an epoch-dependent quantities

$$\rho_1 = 1, \rho_m = \sqrt{\frac{\eta \tau_{m-1}}{\log(\delta^{-1}|\mathcal{F}|m\log(|\mathcal{A}|T))}}, m \geq 2,$$

so $\gamma_t = \sqrt{|S_t|}\rho_{m(t)}$ for $m(t) \geq 2$.

At the end of this subsection, we prove Lemma 3.1:

*Proof.*

- Due to the algorithmic construction, the set $D_t$ found by the greedy algorithm is an independence set of $G_r$, because there does exists connected edge in $D_t$. Hence, the size of $D_t$ is upper bounded by $\alpha(G_t)$. Since the empirically best arm must be selected into $D_t$ and the adjustment of $D_t$ does not change the size of $S_t$, the size of $S_t$ is equal to $|D_t|$, which is bounded by $\alpha(G_t)$.

- Due the algorithmic construction, the LP finds a set $D_t$ with its size

$$\mathbb{E}_{alg}[|D_t|] = \sum_{a \in \mathcal{A}} \mathbb{E}_{alg}[Bernoulli(z_a)] = \delta_f(G_t).$$

  The following adjustment only add new empirically best arm and modify the arms in place, so

$$\mathbb{E}_{alg}[|S_t|] \leq \mathbb{E}_{alg}[|D_t|] + 1 = \delta_f(G_t) + 1.$$

$\square$

To derive the regret upper bound, we also need to define the following high-probability events. These high-probability events and their variants have been proved in literatures Foster et al. (2018); Simchi-Levi & Xu (2021); Foster et al. (2020). These results still hold in graph feedback setting as more samples only help concentration and sampling probability in both options can still cover all actions (in expectation). To avoid repetition, we directly write down the following events:

$$\Gamma = \left\{ \forall m \geq 2, \frac{1}{\tau_{m-1}} \sum_{t=1}^{\tau_{m-1}} \mathbb{E}_{x_t, a_t}\left[ \sum_{a \in \mathcal{N}_{a_t}(G_t)} (\hat{f}_{m(t)}(x_t, a) - f^*(x_t, a))^2 \Big| \mathcal{H}_{t-1} \right] \leq \frac{1}{\rho_m^2} \right\}.$$

Since the data size up to round $t$ is at most $|\mathcal{A}|t$, it is straightforward to show $\Gamma$ holds with probability at least $1 - \delta/2$. This is the result of the union bound and the property of the **Least Square Oracle** that is independent of algorithm design.

## B.2 IMPLICIT OPTIMIZATION PROBLEM

**Lemma B.1** *(Implicit Optimization Problem). For all epoch $m$ and all rounds $t$ in epoch $m$, $Q_t$ is a feasible solution to the following implicit optimization problem:*

$$\sum_{\pi \in \Psi} Q_t(\pi)\widehat{Reg}_t(\pi) \leq \sqrt{|S_t| - 1}/\rho_m \tag{3}$$

$$\mathbb{E}_{x_t}\left[ \frac{1}{p_t(\pi(x_t)|x_t, S_t)} \right] \leq |S_t| + \sqrt{|S_t|}\rho_m\widehat{Reg}_t(\pi), \forall \pi \in \Psi(S_t). \tag{4}$$

*Proof.* Let $m$ and $t$ in epoch $m$ be fixed. We have

$$\sum_{\pi \in \Psi} Q_t(\pi)\widehat{Reg}_t(\pi)$$

$$= \sum_{\pi \in \Psi} Q_t(\pi)\mathbb{E}_{x_t}\left[(\hat{f}_m(x_t, \pi_{\hat{f}_m}(x_t)) - \hat{f}_m(x_t, \pi(x_t)))\right]$$

$$= \mathbb{E}_{x_t}\left[\sum_{a \in \mathcal{A}} \sum_{\pi \in \Psi} \mathbf{I}\{\pi(x_t) = a\} Q_t(\pi)(\hat{f}_m(x_t, \pi_{\hat{f}_m}(x_t)) - \hat{f}_m(x_t, a))\right]$$

$$= \mathbb{E}_{x_t}\left[\sum_{a \in \mathcal{A}} p_t(a|x_t, S_t)(\hat{f}_m(x_t, \pi_{\hat{f}_m}(x_t)) - \hat{f}_m(x_t, a))\right].$$

The first and second equalities are the definitions of $\widehat{Reg}_t(\pi)$ and $Q_t(\pi)$, respectively.

Now for the context $x_t$, we have

$$\sum_{a \in \mathcal{A}} p_t(a|x_t, S_t)(\hat{f}_m(x_t, \pi_{\hat{f}_m}(x_t)) - \hat{f}_m(x_t, a))$$

$$= \sum_{a \in S_t - \{\pi_{\hat{f}_m}(x_t)\}} \frac{\hat{f}_m(x_t, \pi_{\hat{f}_m}(x_t)) - \hat{f}_m(x_t, a)}{|S_t| + \gamma_t(\hat{f}_m(x_t, \pi_{\hat{f}_m}(x_t)) - \hat{f}_m(x_t, a))}$$

$$\leq [|S_t| - 1]/\gamma_t$$

$$\leq \sqrt{|S_t| - 1}/\rho_m.$$

The first equality is due to the construction of $p(\cdot|\cdot)$, which is zero of actions outside $S_t$. Taking expectation over the randomness of $x_t$ and $G_t$, we have

$$\sum_{\pi \in \Psi} Q_t(\pi)\widehat{Reg}_t(\pi) \leq \sqrt{|S_t| - 1}/\rho_m.$$

The first inequality is Jensen's inequality and the second one is due to Lemma 3.1 and the property of the conditional expectation.

For the second inequality, we first observe that for any policy $\pi \in \Psi(S_t)$, given any context $x_t \in \mathcal{X}$,

$$\frac{1}{p_t(\pi(x_t)|x_t, S_t)} = |S_t| + \gamma_t(\hat{f}_m(x_t, \pi_{\hat{f}_m}(x_t)) - \hat{f}_m(x_t, \pi(x_t))),$$

if $\pi(x_t) \neq \pi_{\hat{f}_m}(x_t)$, and

$$\frac{1}{p_t(\pi(x_t)|x_t, S_t)} \leq \frac{1}{1/|S_t|} = |S_t| + \gamma_t(\hat{f}_m(x_t, \pi_{\hat{f}_m}(x_t)) - \hat{f}_m(x_t, \pi(x_t))),$$

if $\pi(x_t) = \pi_{\hat{f}_m}(x_t)$. The result follows immediately by taking expectation over $x_t$ and $G_t$. $\square$

Compared with IOP in Simchi-Levi & Xu (2021), the key different part is that $|S_t|$ is replaced by the cardinality $|\mathcal{A}|$ of the whole action set. Another difference lies in the universal policy space $\Psi$. Since we only consider a subspace of $\Psi$, it will be possible for us to reduce the dependence of action sizes in regrets. These two points highlight the adaptivity to graph feedbacks and show how the graph structure affects the action selection.

### B.3 PREDICTION ERROR

Our setting do not change the proof procedure of the following lemma Simchi-Levi & Xu (2021), because this lemma does not explicitly involve the number of action set. This lemma bounds the prediction error between the true reward and the estimated reward.

**Lemma B.2** *Assume $\Gamma$ holds. For all epochs $m > 1$, all rounds $t$ in epoch $m$, and all policies $\pi \in \Psi(S_t)$, then*

$$\left|\widehat{\mathcal{R}}_t(\pi) - \mathcal{R}_t(\pi)\right| \leq \frac{1}{2\rho_m}\sqrt{\max_{1 \leq s \leq \tau_{m(t)-1}} \mathbb{E}\left[\frac{1}{p_s(\pi(x_s)|x_s, S_s)}\right]},$$

*where the expectation is taken with respect to the randomness of $x_s$ and $S_s$.*

*Proof.* For any fixed round $t$ and any policy $\pi \in \Psi(S_t)$, we have

$$\widehat{\mathcal{R}}_t(\pi) - \mathcal{R}_t(\pi) = \mathbb{E}_{x_t}[\hat{f}_{m(t)}(x_t, \pi(x_t)) - f^*(x_t, \pi(x_t))].$$

For all $s = 1, 2, \cdots, \tau_{m(t)-1}$, we have

$$\mathbb{E}_{a_s|x_s}\left[\sum_{a \in \mathcal{N}_{a_s}(G_s)} (\hat{f}_{m(t)}(x_s, a) - f^*(x_s, a))^2 \middle| \mathcal{H}_{s-1}, S_s\right]$$

$$= \sum_{a \in \mathcal{A}} p_s(a|x_s, S_s) \sum_{a' \in \mathcal{N}_a(G_s)} (\hat{f}_{m(t)}(x_s, a') - f^*(x_s, a'))^2$$

$$\geq p_s(\pi(x_s)|x_s, S_s)(\hat{f}_{m(t)}(x_s, \pi(x_s)) - f^*(x_s, \pi(x_s)))^2.$$

Specifically, we have

$$\mathbb{E}_{a_s|x_s}\left[\sum_{a \in \mathcal{N}_{a_s}(G_s)} (\hat{f}_{m(t)}(x_s, a) - f^*(x_s, a))^2 \middle| \mathcal{H}_{s-1}\right]$$

$$\geq \mathbb{E}_{S_s}[p_s(\pi(x_s)|x_s, S_s)](\hat{f}_{m(t)}(x_s, \pi(x_s)) - f^*(x_s, \pi(x_s)))^2.$$

Therefore, we have

$$\max_{1 \leq s \leq \tau_{m(t)-1}} \mathbb{E}\left[\frac{1}{p_s(\pi(x_s)|x_s, S_s)}\right] \sum_{s=1}^{\tau_{m(t)-1}} \mathbb{E}_{a_s,x_s}\left[\sum_{a \in \mathcal{N}_{a_s}(G_s)} (\hat{f}_{m(t)}(x_s, a) - f^*(x_s, a))^2 \middle| \mathcal{H}_{s-1}\right]$$

$$= \sum_{s=1}^{\tau_{m(t)-1}} \mathbb{E}\left[\frac{1}{p_s(\pi(x_s)|x_s, S_s)}\right] \mathbb{E}_{a_s,x_s}\left[\sum_{a \in \mathcal{N}_{a_s}(G_s)} (\hat{f}_{m(t)}(x_s, a) - f^*(x_s, a))^2 \middle| \mathcal{H}_{s-1}\right]$$

$$= \sum_{s=1}^{\tau_{m(t)-1}} \mathbb{E}_{x_s}\mathbb{E}_{S_s}\left[\frac{1}{p_s(\pi(x_s)|x_s, S_s)}\right] \mathbb{E}_{x_s}\mathbb{E}_{a_s|x_s}\left[\sum_{a \in \mathcal{N}_{a_s}(G_s)} (\hat{f}_{m(t)}(x_s, a) - f^*(x_s, a))^2 \middle| \mathcal{H}_{s-1}\right]$$

$$\geq \sum_{s=1}^{\tau_{m(t)-1}} \left(\mathbb{E}_{x_s}\left[\sqrt{\mathbb{E}_{S_s}\left[\frac{1}{p_s(\pi(x_s)|x_s, S_s)}\right]\mathbb{E}_{a_s|x_s}\left[\sum_{a \in \mathcal{N}_{a_s}(G_s)} (\hat{f}_{m(t)}(x_s, a) - f^*(x_s, a))^2 \middle| \mathcal{H}_{s-1}\right]}\right]\right)^2$$

$$\geq \sum_{s=1}^{\tau_{m(t)-1}} \left(\mathbb{E}_{x_s}\left[\sqrt{\mathbb{E}_{S_s}\left[\frac{1}{p_s(\pi(x_s)|x_s, S_s)}\right]\mathbb{E}_{S_s}\left[p_s(\pi(x_s)|x_s, S_s)\right](\hat{f}_{m(t)}(x_s, \pi(x_s)) - f^*(x_s, \pi(x_s)))^2}\right]\right)^2$$

$$\geq \sum_{s=1}^{\tau_{m(t)-1}} \left(\mathbb{E}_{x_s}[|\hat{f}_{m(t)}(x_s, \pi(x_s)) - f^*(x_s, \pi(x_s))|]\right)^2$$

$$\geq \sum_{s=1}^{\tau_{m(t)-1}} |\widehat{\mathcal{R}}_t(\pi) - \mathcal{R}_t(\pi)|^2$$

$$= \tau_{m(t)-1}|\widehat{\mathcal{R}}_t(\pi) - \mathcal{R}_t(\pi)|^2.$$

Here, the inequalities result from Cauchy-Schwarz inequality, the previous deduction, the Jensen's inequality and the convexity of $L_1$ norm, respectively. The final equality results from the i.i.d. assumption on context distribution. Therefore,

$$|\widehat{\mathcal{R}}_t(\pi) - \mathcal{R}_t(\pi)|$$

$$\leq \sqrt{\max_{1 \leq s \leq \tau_{m(t)-1}} \mathbb{E}\left[\frac{1}{p_s(\pi(x_s)|x_s, S_s)}\right] \frac{1}{\tau_{m(t)-1}} \sum_{s=1}^{\tau_{m(t)-1}} \mathbb{E}_{a_s,x_s}\left[\sum_{a \in \mathcal{N}_{a_s}(G_s)} (\hat{f}_{m(t)}(x_s, a) - f^*(x_s, a))^2 \middle| \mathcal{H}_{s-1}\right]}.$$

We conclude the proof by plugging in the definition of $\Gamma$. $\qquad\square$

The third step is to show that the one-step regret $Reg_t(\pi)$ is close to the one-step estimated regret $\widehat{Reg}_t(\pi)$. In the following lemma, we mainly focus on the first option. Thanks to Lemma 3.1, the following results can be easily extended to the second option.

**Lemma B.3** *Assume $\Gamma$ holds. Let $c_0 = 4$. For all epochs $m$ and all rounds $t$ in epoch $m$, and all policies $\pi \in \Psi(S_t)$,*

$$Reg(\pi) \leq 2\widehat{Reg}_t(\pi) + c_0\sqrt{\mathbb{E}_G[\alpha(G)]}/\rho_m, \tag{5}$$

$$\widehat{Reg}_t(\pi) \leq 2Reg(\pi) + c_0\sqrt{\mathbb{E}_G[\alpha(G)]}/\rho_m. \tag{6}$$

*Proof.* We prove this lemma via induction on $m$. It is easy to check

$$Reg(\pi) \leq 1, \widehat{Reg}_t(\pi) \leq 1,$$

as $\gamma_1 = 0$ and $c_0\mathbb{E}_G\big[\alpha(G_t)\big] \geq 1$. Hence, the base case holds.

For the inductive step, fix some epoch $m > 1$ and assume that for all epochs $m' < m$, all rounds $t'$ in epoch $m'$, and all $\pi \in \Psi$, the inequalities (9) and (10) hold. We first show that for all rounds $t$ in epoch $m$ and all $\pi \in \Psi$,

$$Reg(\pi) \leq 2\widehat{Reg}_t(\pi) + c_0\sqrt{\mathbb{E}_G[\alpha(G)]}/\rho_m.$$

We have

$$Reg(\pi) - \widehat{Reg}_t(\pi)$$

$$= [\mathcal{R}(\pi_{f^*}) - \mathcal{R}(\pi)] - [\widehat{\mathcal{R}}_t(\pi_{\hat{f}_m}) - \widehat{\mathcal{R}}_t(\pi)]$$

$$\leq [\mathcal{R}(\pi_{f^*}) - \mathcal{R}(\pi)] - [\widehat{\mathcal{R}}_t(\pi_{f^*}) - \widehat{\mathcal{R}}_t(\pi)]$$

$$\leq |\mathcal{R}(\pi_{f^*}) - \widehat{\mathcal{R}}_t(\pi_{f^*})| + |\mathcal{R}(\pi) - \widehat{\mathcal{R}}_t(\pi)|$$

$$\leq \frac{1}{2\rho_m}\sqrt{\max_{1 \leq s \leq \tau_{m(t)-1}} \mathbb{E}\left[\frac{1}{p_s(\pi_{f^*}(x_s)|x_s, S_s)}\right]} + \frac{1}{2\rho_m}\sqrt{\max_{1 \leq s \leq \tau_{m(t)-1}} \mathbb{E}\left[\frac{1}{p_s(\pi(x_s)|x_s, S_s)}\right]}$$

$$\leq \frac{\max_{1 \leq s \leq \tau_{m(t)-1}} \mathbb{E}\left[\frac{1}{p_s(\pi_{f^*}(x_s)|x_s, S_s)}\right]}{5\rho_m\sqrt{\mathbb{E}_G[\alpha(G)]}} + \frac{\max_{1 \leq s \leq \tau_{m(t)-1}} \mathbb{E}\left[\frac{1}{p_s(\pi(x_s)|x_s, S_s)}\right]}{5\rho_m\sqrt{\mathbb{E}_G[\alpha(G)]}} + \frac{5\sqrt{\mathbb{E}_G[\alpha(G)]}}{8\rho_m}.$$

The last inequality is by the AM-GM inequality.

From Lemma B.1 and Lemma 3.1 we know that

$$\max_{1 \leq s \leq \tau_{m(t)-1}} \mathbb{E}\left[\frac{1}{p_s(\pi_{f^*}(x_s)|x_s, S_s)}\right] \leq \mathbb{E}_G[\alpha(G)] + \sqrt{\mathbb{E}_G[\alpha(G)]}\rho_m\widehat{Reg}_t(\pi),$$

holds for all $\pi \in \Psi$, for all epoch $m \in [M]$ and for all rounds $t$ in corresponding epochs. Hence, for epoch $m$ and all rounds $t$ in this epoch, we have

$$\frac{\max_{1 \leq s \leq \tau_{m(t)-1}} \mathbb{E}\left[\frac{1}{p_s(\pi(x_s)|x_s, S_s)}\right]}{5\rho_m\sqrt{\mathbb{E}_G[\alpha(G)]}}$$

$$\leq \frac{\mathbb{E}_G[\alpha(G)] + \mathbb{E}_G[\sqrt{\alpha(G)}]\rho_{m-1}\widehat{Reg}_t(\pi)}{5\sqrt{\mathbb{E}_G[\alpha(G)]}\rho_m}, \text{ (Lemma B.1)}$$

$$\leq \frac{\mathbb{E}_G[\alpha(G)] + \mathbb{E}_G[\sqrt{\alpha(G)}]\rho_{m-1}[2Reg(\pi) + c_0\sqrt{\mathbb{E}_G[\alpha(G)]}/\rho_{m-1}]}{5\sqrt{\mathbb{E}_G[\alpha(G)]}\rho_m}, \text{ (inductive assumption)}$$

$$\leq \frac{\mathbb{E}_G[\alpha(G)] + \sqrt{\mathbb{E}_G[\alpha(G)]}\rho_{m-1}[2Reg(\pi) + c_0\sqrt{\mathbb{E}_G[\alpha(G)]}/\rho_{m-1}]}{5\sqrt{\mathbb{E}_G[\alpha(G)]}\rho_m}, \text{ (Jensen's inequality)}$$

$$\leq \frac{2}{5}Reg(\pi)\frac{\rho_{m-1}}{\rho_m} + \frac{1 + c_0}{5\rho_m}\sqrt{\mathbb{E}_G[\alpha(G)]}.$$

We can bound $\dfrac{\max_{1 \le s \le \tau_{m(t)-1}} \mathbb{E}\left[\frac{1}{p_s(\pi_{f^*}(x_s)|x_s, S_s)}\right]}{5\rho_m \sqrt{\mathbb{E}_G[\alpha(G)]}}$ in the same way.

Combing all above inequalities yields

$$
\begin{aligned}
Reg(\pi) - \widehat{Reg}_t(\pi) \le & \frac{2(1+c_0)\sqrt{\mathbb{E}_G[\alpha(G)]}}{5\rho_m} + \frac{4}{5}\widehat{Reg}_t(\pi) + \frac{5\sqrt{\mathbb{E}_G[\alpha(G)]}}{8\rho_m} \\
\le & \widehat{Reg}_t(\pi) + (2(1+\frac{c_0}{5}) + \frac{5}{8})\frac{\sqrt{\mathbb{E}_G[\alpha(G)]}}{\rho_m} \\
\le & \widehat{Reg}_t(\pi) + c_0\frac{\sqrt{\mathbb{E}_G[\alpha(G)]}}{\rho_m}.
\end{aligned}
$$

Similarly, we have

$$
\begin{aligned}
& \widehat{Reg}_t(\pi) - Reg(\pi) \\
= & [\widehat{\mathcal{R}}_t(\pi_{\hat{f}_m}) - \widehat{\mathcal{R}}_t(\pi)] - [\mathcal{R}(\pi_{f^*}) - \mathcal{R}(\pi)] \\
\le & [\widehat{\mathcal{R}}_t(\pi_{\hat{f}_m}) - \widehat{\mathcal{R}}_t(\pi)] - [\mathcal{R}(\pi_{\hat{f}_m}) - \mathcal{R}(\pi)] \\
\le & |\mathcal{R}(\pi_{\hat{f}_m}) - \widehat{\mathcal{R}}_t(\pi_{\hat{f}_m})| + |\mathcal{R}(\pi) - \widehat{\mathcal{R}}_t(\pi)|.
\end{aligned}
$$

We can bound the above terms in the same steps.

$\square$

Now it is time to prove the theorem of the minimax regret.

*Proof.* Our regret analysis builds on the framework in Simchi-Levi & Xu (2021).

**Step 1:** proving an implicit optimization problem for $Q_t$ in Lemma B.1.

**Step 2:** bounding the prediction error between $\widehat{\mathcal{R}}_t(\pi)$ and $\mathcal{R}_t(\pi)$ in Lemma B.2. Then we can show that the one-step regrets $\widehat{Reg}_t(\pi)$ and $Reg(\pi)$ are close to each other.

**Step 3:** bounding the cumulative regret $Reg(T)$.

By Lemma 4 of Simchi-Levi & Xu (2021),

$$
\mathbb{E}[Reg(T)] = \sum_{t=1}^T \sum_{\pi \in \Psi} Q_t(\pi)Reg(\pi) = \sum_{t=1}^T \sum_{\pi \in \Psi(S_t)} Q_t(\pi)Reg(\pi).
$$

At the round $t$, the policy space $\Psi$ shrinks to $\Psi(S_t)$ so we only need to consider the polices in this subspace. From Lemma B.3, we know

$$
Reg(\pi) \le 2\widehat{Reg}_t(\pi) + c_0\sqrt{\mathbb{E}_G[\alpha(G)]}/\rho_m
$$

so

$$\mathbb{E}[Reg(T)] = \sum_{t=1}^{T} \sum_{\pi \in \Psi(S_t)} Q_t(\pi) Reg(\pi)$$

$$\leq 2 \sum_{t=1}^{T} \sum_{\pi \in \Psi(S_t)} Q_t(\pi) \widehat{Reg}_t(\pi) + \sum_{t=1}^{T} c_0 \sqrt{\mathbb{E}_G[\alpha(G)]} / \rho_{m(t)}$$

$$\leq (2 + c_0) \sqrt{\mathbb{E}_G[\alpha(G)]} \sum_{t=1}^{T} \frac{1}{\rho_{m(t)}}$$

$$\leq (2 + c_0) \sqrt{\mathbb{E}_G[\alpha(G)]} \sum_{m=1}^{\lceil \log_2 T \rceil} \sqrt{\log(2\delta^{-1}|\mathcal{F}| \log^2 T) \tau_{m-1} / \eta}$$

$$\leq (2 + c_0) \sqrt{\mathbb{E}_G[\alpha(G)] \log(2\delta^{-1}|\mathcal{F}| \log^2 T)/\eta} \sum_{m=1}^{\lceil \log_2 T \rceil} \frac{\tau_m - \tau_{m-1}}{\sqrt{\tau_{m-1}}}$$

$$\leq (2 + c_0) \sqrt{\mathbb{E}_G[\alpha(G)] \log(2\delta^{-1}|\mathcal{F}| \log^2 T)/\eta} \sum_{m=1}^{\lceil \log_2 T \rceil} \sqrt{2}^{m-1}$$

$$\leq 17.875 \sqrt{\mathbb{E}_G[\alpha(G)] \log(2\delta^{-1}|\mathcal{F}| \log^2 T) T/\eta}.$$

$\square$

We have completed the proof of the first option. For the second option, we only need to plug in the expected upper bound $\mathbb{E}_G[\delta_f(G)]$. The remaining steps do not change so we can still obtain the final regret upper bounds.

## B.4 INCORPORATING THE UNIFORM GAP

To incorporate the instance gaps, we define the following quantities that are used to show the regret upper bound:

$$q_m = \mathbb{E}_x[|\mathcal{A}(x; \mathcal{F}_m)| > 1]$$
$$\hat{q}_m = \mathbb{E}_{x \sim \mathcal{D}_m}[|\mathcal{A}(x; \mathcal{F}_m)| > 1]$$
$$w_m = q_m + \mu_m$$
$$\hat{w}_m = \hat{q}_m + \mu_m.$$

Similarly, we need to define the following high-probability event $\Gamma$.

**Lemma B.4 (Foster et al. (2018; 2020))** *Let $C_\delta = 16 \log(2\delta^{-1}|\mathcal{F}||\mathcal{A}|^2 T^2)$ and $M = \lceil \log T \rceil$. Denote the following events as $\Gamma_i, i = 1, 2, 3$, respectively.*

1. *For all $m \in [M]$, for all $\beta_m \geq 0$,*

$$\sum_{t=\tau_{m-2}}^{\tau_{m-1}} \mathbb{E}_{x_t, a_t}[\sum_{a \in \mathcal{N}_{a_t}(G_t)} (\hat{f}_m(x_t, a) - f^*(x_t, a))^2 | \mathcal{H}_{t-1}] \leq C_\delta$$

*and*

$$\forall f \in \mathcal{F}_m, \sum_{t=\tau_{m-2}}^{\tau_{m-1}} \mathbb{E}_{x_t, a_t}[\sum_{a \in \mathcal{N}_{a_t}(G_t)} (\hat{f}_m(x_t, a) - f^*(x_t, a))^2 | \mathcal{H}_{t-1}] \leq 2\beta_m + C_\delta.$$

2. *For all $m \in [M]$, we have $f^* \in \mathcal{F}_M \subset \mathcal{F}_{M-1} \subset \cdots \mathcal{F}_1$.*

3. *For all $m \in [M]$, we have $\frac{2}{3} w_m \leq \frac{2}{3} \hat{w}_m \leq \frac{2}{3} w_m$.*

*Then with probability $1 - \delta$, the event $\Gamma = \bigcap_{i=1}^{3} \Gamma_i$ holds.*

Given the high-probability event $\Gamma$, we observe $\hat{f}_m \in \mathcal{F}_m$ and $f^* \in \mathcal{F}_m$ for all $m \in [M]$. Since $|S_t| = 1$, we must have $\pi_f(x) = \pi_{f^*}(x) = \pi_{\hat{f}_m}(x)$. Therefore, we only need to consider $\mathcal{R}_t^{dis}(\pi)$ and $\hat{\mathcal{R}}_t^{dis}(\pi)$. Define the following quantities:

$$\mathcal{R}_t^{dis}(\pi) = \mathbb{E}[\mathbf{I}\{|S_t| > 1\} f^*(x, \pi(x))] \text{ and } \hat{\mathcal{R}}_t^{dis}(\pi) = \mathbb{E}[\mathbf{I}\{|S_t| > 1\} \hat{f}_m(x, \pi(x))].$$

We conclude that

$$Reg_t(\pi) = \mathcal{R}_t^{dis}(\pi_{f^*}) - \mathcal{R}_t^{dis}(\pi)$$

and

$$\widehat{Reg}_t(\pi) = \widehat{\mathcal{R}}_t^{dis}(\pi_{f^*}) - \widehat{\mathcal{R}}_t^{dis}(\pi)$$

when $\Gamma$ holds. Now we prove a proposition of selected quantities $\lambda_m = \hat{w}_m / \sqrt{\hat{w}_{m-1}}$.

**Proposition B.1** *Assume that $\Gamma$ holds. Then $\rho_m / \hat{w}_m$ is monotonically non-decreasing in $m$.*

*Proof.* We have $\frac{\rho_1}{\hat{w}_1} = 0$ for $m = 1$ and

$$\frac{\rho_m}{\hat{w}_m} = \sqrt{\frac{\eta(\tau_{m-1} - \tau_{m-2})}{\hat{w}_m \log(2\delta^{-1}|\mathcal{F}||\mathcal{A}|^2 T^2)}}$$

for all $m = 2, \cdots, M$.

Clearly, the property holds for $m = 2$. Hence, for all $m = 3, \cdots, M$, we have

$$\frac{\rho_{m-1}}{\hat{w}_{m-1}} \Big/ \frac{\rho_m}{\hat{w}_m} = \sqrt{\frac{\hat{w}_{m-2}}{\hat{w}_{m-1}} \frac{(\tau_{m-2} - \tau_{m-3})}{(\tau_{m-1} - \tau_{m-2})}} = \sqrt{\frac{\hat{w}_{m-2}}{2\hat{w}_{m-1}}} \leq \sqrt{\frac{\frac{4}{3} w_{m-2}}{2 \times \frac{2}{3} w_{m-1}}} = \sqrt{\frac{w_{m-2}}{w_{m-1}}} \leq 1.$$

The last inequality due to the fact that $\mathcal{F}_M \subset \mathcal{F}_{M-1} \cdots \mathcal{F}_1$ and thus $w_{m-2} = q_{m-2} + \mu_{m-2} \leq q_{m-1} + \mu_{m-1} \leq w_{m-1}$.

$\square$

Then we need to modify the proof of Lemma B.1 to incorporate the quantity $q_m$. We mainly focus on the first option.

**Corollary B.1** *(Disagreement-based Implicit Optimization Problem). For all epoch $m$ and all rounds $t$ in epoch $m$, $Q_t$ is a feasible solution to the following implicit optimization problem:*

$$\mathbb{E}\Big[\sum_{\pi \in \Psi} Q_t(\pi) \widehat{Reg}_t(\pi)\Big] \leq q_m \sqrt{\mathbb{E}_G[\alpha(G)]} / \rho_m \tag{7}$$

$$\mathbb{E}\left[\frac{\mathbf{I}\{|S_t| > 1\}}{p_t(\pi(x_t)|x_t, S_t)}\right] \leq q_m \mathbb{E}_G[\alpha(G)] + \sqrt{\mathbb{E}_G[\alpha(G)]} \rho_m \widehat{Reg}_t(\pi), \forall \pi \in \Psi(S_t). \tag{8}$$

*Proof.* From the proof of Lemma B.1, we know that

$$\mathbb{E}\Big[\sum_{\pi \in \Psi} Q_t(\pi) \widehat{Reg}_t(\pi)\Big]$$

$$\leq \mathbb{E}[\sqrt{|S_t| - 1}] / \rho_m$$

$$\leq (\mathbb{E}[\sqrt{|S_t| - 1}\mathbf{I}\{|S_t| > 1\}] + \mathbb{E}[\sqrt{|S_t| - 1}\mathbf{I}\{|S_t| = 1\}]) / \rho_m$$

$$\leq \mathbb{E}[\sqrt{|S_t| - 1}\mathbf{I}\{|S_t| > 1\}] / \rho_m$$

$$\leq \mathbb{E}[\sqrt{|S_t| - 1}\mathbf{I}\{|\mathcal{A}(x_t; \mathcal{F}_m)| > 1\}] / \rho_m$$

$$= \mathbb{E}[\sqrt{|S_t| - 1}\mathbb{E}[\mathbf{I}\{|\mathcal{A}(x_t; \mathcal{F}_m)| > 1\}|G_t] / \rho_m$$

$$\leq q_m \sqrt{\mathbb{E}_G[\alpha(G)] - 1} / \rho_m.$$

The fourth inequality is due to the fact that $\mathbf{I}\{|S_t| > 1\}$ implies $\mathbf{I}\{|\mathcal{A}(x_t; \mathcal{F}_m)| > 1\}$. The last equality is due to the independence of $x_t$ and $G_t$.

For the second inequality, we have

$$
\begin{aligned}
&\mathbb{E}\left[\frac{\mathbf{I}\{|S_t| > 1\}}{p_t(\pi(x_t)|x_t, S_t)}\right] \\
=&\mathbb{E}\left[\frac{1 - \mathbf{I}\{|S_t| = 1\}}{p_t(\pi(x_t)|x_t, S_t)}\right] \\
=&\mathbb{E}\left[\frac{1}{p_t(\pi(x_t)|x_t, S_t)}\right] - \mathbb{E}\left[\frac{\mathbf{I}\{|S_t| = 1\}}{p_t(\pi(x_t)|x_t, S_t)}\right] \\
=&\mathbb{E}\left[\frac{1}{p_t(\pi(x_t)|x_t, S_t)}\right] - \mathbb{P}(|S_t| = 1) \\
\leq&\mathbb{E}_{S_t}[|S_t|] + \sqrt{\mathbb{E}_G[\alpha(G)]}\rho_m \widehat{Reg}_t(\pi) - \mathbb{P}(|S_t| = 1) \\
\leq&\mathbb{E}_{S_t}[|S_t|\mathbf{I}\{|S_t| = 1\}] + \mathbb{E}_{S_t}[|S_t|\mathbf{I}\{|S_t| > 1\}] + \sqrt{\mathbb{E}_G[\alpha(G)]}\rho_m \widehat{Reg}_t(\pi) - \mathbb{P}(|S_t| = 1) \\
\leq& q_m \mathbb{E}_G[\alpha(G)] + \sqrt{\mathbb{E}_G[\alpha(G)]}\rho_m \widehat{Reg}_t(\pi).
\end{aligned}
$$

$\square$

For the second step, we directly write down the results as the proof does not involve any action sizes. We can follow the similar proof steps as Lemma B.2.

**Lemma B.5** *Assume $\Gamma$ holds. For all epochs $m > 1$, all rounds $t$ in epoch $m$, and all policies $\pi \in \Psi(S_t)$, then*

$$
\left|\widehat{\mathcal{R}}_t(\pi) - \mathcal{R}_t(\pi)\right| \leq \frac{\lambda_m}{2\rho_m}\sqrt{\max_{1 \leq s \leq \tau_{m(t)}-1} \mathbb{E}\left[\frac{\mathbf{I}\{|S_s| > 1\}}{p_s(\pi(x_s)|x_s, S_s)}\right]},
$$

*where the expectation is taken with respect to the randomness of $x_s$ and $S_s$.*

The third step is to show that the one-step regret $Reg_t(\pi)$ is close to the one-step estimated regret $\widehat{Reg}_t(\pi)$. The following lemma states the result.

**Lemma B.6** *Assume $\Gamma$ holds. Let $c_0 = 4$. For all epochs $m$ and all rounds $t$ in epoch $m$, and all policies $\pi \in \Psi(S_t)$,*

$$
Reg_t(\pi) \leq 2\widehat{Reg}_t(\pi) + c_0 \hat{w}_m \sqrt{\mathbb{E}_G[\alpha(G)]}/\rho_m, \tag{9}
$$

$$
\widehat{Reg}_t(\pi) \leq 2Reg_t(\pi) + c_0 \hat{w}_m \sqrt{\mathbb{E}_G[\alpha(G)]}/\rho_m. \tag{10}
$$

*Proof.* We prove this lemma via induction on $m$. It is easy to check

$$
Reg_t(\pi) \leq 1, \widehat{Reg}_t(\pi) \leq 1,
$$

as $\gamma_1 = 0$ and $c_0 \mathbb{E}_G[\alpha(G)] \geq 1$. Hence, the base case holds.

For the inductive step, fix some epoch $m > 1$ and assume that for all epochs $m' < m$, all rounds $t'$ in epoch $m'$, and all $\pi \in \Psi$, the inequalities (9) and (10) hold. We first show that for all rounds $t$ in epoch $m$ and all $\pi \in \Psi$,

$$
Reg_t(\pi) \leq 2\widehat{Reg}_t(\pi) + c_0 \hat{w}_m \sqrt{\mathbb{E}_G[\alpha(G_t)]}/\rho_m.
$$

We have

$$Reg_t(\pi) - \widehat{Reg}_t(\pi)$$

$$=[\mathcal{R}^{dis}(\pi_{f^*}) - \mathcal{R}^{dis}(\pi)] - [\widehat{\mathcal{R}}_t^{dis}(\pi_{\hat{f}_m}) - \widehat{\mathcal{R}}_t^{dis}(\pi)]$$

$$\leq[\mathcal{R}^{dis}(\pi_{f^*}) - \mathcal{R}^{dis}(\pi)] - [\widehat{\mathcal{R}}_t^{dis}(\pi_{f^*}) - \widehat{\mathcal{R}}_t^{dis}(\pi)]$$

$$\leq|\mathcal{R}^{dis}(\pi_{f^*}) - \widehat{\mathcal{R}}_t^{dis}(\pi_{f^*})| + |\mathcal{R}^{dis}(\pi) - \widehat{\mathcal{R}}_t^{dis}(\pi)|$$

$$\leq\frac{\lambda_m}{2\rho_m}\sqrt{\max_{1\leq s\leq\tau_{m(t)-1}}\mathbb{E}\left[\frac{\mathbf{I}\{|S_s| > 1\}}{p_s(\pi_{f^*}(x_s)|x_s, S_s)}\right]} + \frac{\lambda_m}{2\rho_m}\sqrt{\max_{1\leq s\leq\tau_{m(t)-1}}\mathbb{E}\left[\frac{\mathbf{I}\{|S_s| > 1\}}{p_s(\pi(x_s)|x_s, S_s)}\right]}$$

$$\leq\frac{\max_{1\leq s\leq\tau_{m(t)-1}}\mathbb{E}\left[\frac{\mathbf{I}\{|S_s|>1\}}{p_s(\pi_{f^*}(x_s)|x_s,S_s)}\right]}{5\rho_m\sqrt{\mathbb{E}_G[\alpha(G)]\hat{w}_{m-1}/\hat{w}_m}} + \frac{\max_{1\leq s\leq\tau_{m(t)-1}}\mathbb{E}\left[\frac{\mathbf{I}\{|S_s|>1\}}{p_s(\pi(x_s)|x_s,S_s)}\right]}{5\rho_m\sqrt{\mathbb{E}_G[\alpha(G)]\hat{w}_{m-1}/\hat{w}_m}} + \frac{5\sqrt{\mathbb{E}_G[\alpha(G)]}\hat{w}_m}{8\rho_m}.$$

The last inequality is by the AM-GM inequality.

From Corollary B.1 we know that

$$\max_{1\leq s\leq\tau_{m(t)-1}}\mathbb{E}\left[\frac{\mathbf{I}\{|S_s| > 1\}}{p_s(\pi(x_s)|x_s, S_s)}\right] \leq q_{m-1}\mathbb{E}_G[\alpha(G)] + \mathbb{E}_G[\sqrt{\alpha(G)}]\rho_{m-1}\widehat{Reg}_t(\pi),$$

holds for all $\pi \in \Psi$, for all epoch $m \in [M]$ and for all rounds $t$ in corresponding epochs. Hence, for epoch $m$ and all rounds $t$ in this epoch, we have

$$\frac{\max_{1\leq s\leq\tau_{m(t)-1}}\mathbb{E}\left[\frac{\mathbf{I}\{|S_s|>1\}}{p_s(\pi(x_s)|x_s,S_s)}\right]}{5\rho_m\sqrt{\mathbb{E}_G[\alpha(G)]\hat{w}_{m-1}/\hat{w}_m}}$$

$$\leq\frac{q_{m-1}\mathbb{E}_G[\alpha(G)] + \mathbb{E}_G[\sqrt{\alpha(G)}]\rho_{m-1}\widehat{Reg}_t(\pi)}{5\sqrt{\mathbb{E}_G[\alpha(G)]}\rho_m\hat{w}_{m-1}/\hat{w}_m}, \text{ (Corollary B.1)}$$

$$\leq\frac{q_{m-1}\mathbb{E}_G[\alpha(G)] + \mathbb{E}_G[\sqrt{\alpha(G)}]\rho_{m-1}[2Reg_t(\pi) + c_0\hat{w}_{m-1}\sqrt{\mathbb{E}_G[\alpha(G)]}/\rho_{m-1}]}{5\sqrt{\mathbb{E}_G[\alpha(G)]}\rho_m\hat{w}_{m-1}/\hat{w}_m}, \text{ (inductive assumption)}$$

$$\leq\frac{q_{m-1}\mathbb{E}_G[\alpha(G)] + \sqrt{\mathbb{E}_G[\alpha(G)]}\rho_{m-1}[2Reg_t(\pi) + c_0\hat{w}_{m-1}\sqrt{\mathbb{E}_G[\alpha(G)]}/\rho_{m-1}]}{5\sqrt{\mathbb{E}_G[\alpha(G)]}\rho_m\hat{w}_{m-1}/\hat{w}_m}, \text{ (Jensen's inequality)}$$

$$\leq\frac{2}{5}Reg_t(\pi)\frac{\rho_{m-1}/\hat{w}_{m-1}}{\rho_m/\hat{w}_m} + \frac{q_{m-1} + c_0\hat{w}_{m-1}}{5\rho_m\hat{w}_{m-1}/\hat{w}_m}\sqrt{\mathbb{E}_G[\alpha(G)]}, \text{ (Proposition B.1 and } q_{m-1} \leq w_m)$$

$$\leq\frac{2}{5}Reg_t(\pi) + \frac{4/3 + c_0}{5}\hat{w}_m\sqrt{\mathbb{E}_G[\alpha(G)]}/\rho_m.$$

We can bound $\frac{\max_{1\leq s\leq\tau_{m(t)-1}}\mathbb{E}\left[\frac{\mathbf{I}\{|S_s|>1\}}{p_s(\pi_{f^*}(x_s)|x_s,S_s)}\right]}{5\rho_m\sqrt{\mathbb{E}_G[\alpha(G)]\hat{w}_{m-1}/\hat{w}_m}}$ in the same way.

Combing all above inequalities yields

$$Reg_t(\pi) - \widehat{Reg}_t(\pi) \leq\frac{2(4/3 + c_0)\hat{w}_m\sqrt{\mathbb{E}_G[\alpha(G)]}}{5\rho_m} + \frac{4}{5}\widehat{Reg}_t(\pi) + \frac{5\sqrt{\mathbb{E}_G[\alpha(G)]}\hat{w}_m}{8\rho_m}$$

$$\leq\widehat{Reg}_t(\pi) + (2(\frac{4}{3} + \frac{c_0}{5}) + \frac{5}{8})\hat{w}_m\frac{\sqrt{\mathbb{E}_G[\alpha(G)]}}{\rho_m}$$

$$\leq\widehat{Reg}_t(\pi) + c_0\frac{\hat{w}_m\sqrt{\mathbb{E}_G[\alpha(G)]}}{\rho_m}.$$

Similarly, we have

$$\widehat{Reg}_t(\pi) - Reg_t(\pi)$$

$$=[\widehat{\mathcal{R}}_t^{dis}(\pi_{\hat{f}_m}) - \widehat{\mathcal{R}}_t^{dis}(\pi)] - [\mathcal{R}^{dis}(\pi_{f^*}) - \mathcal{R}^{dis}(\pi)]$$

$$\leq[\widehat{\mathcal{R}}_t^{dis}(\pi_{\hat{f}_m}) - \widehat{\mathcal{R}}_t^{dis}(\pi)] - [\mathcal{R}^{dis}(\pi_{\hat{f}_m}) - \mathcal{R}^{dis}(\pi)]$$

$$\leq|\mathcal{R}^{dis}(\pi_{\hat{f}_m}) - \widehat{\mathcal{R}}_t^{dis}(\pi_{\hat{f}_m})| + |\mathcal{R}^{dis}(\pi) - \widehat{\mathcal{R}}_t^{dis}(\pi)|.$$

We can bound the above terms in the same steps.

$\square$

**Lemma B.7** *Assume $\Gamma$ holds. For all epochs $m$ and all rounds $t$ in epoch $m$, and all $f \in \mathcal{F}_m$, we have*
$$Reg_t(\pi_f) \le 6\hat{w}_m\sqrt{\mathbb{E}_G[\alpha(G)]}M/\rho_m.$$

*Proof.* We rewrite $Reg_t(\pi_f)$ as $\mathbb{E}[\mathbf{I}\{\pi_f(x) \ne \pi_{f^*}\}(f^*(x, \pi_{f^*}(x)) - f^*(x, \pi_f(x)))]$. Hence we have

$$
\begin{aligned}
&Reg_t(\pi_f) \\
=&\mathbb{E}[\mathbf{I}\{\pi_f(x) \ne \pi_{f^*}(x)\}(f^*(x, \pi_{f^*}(x)) - f^*(x, \pi_f(x)))] \\
=&\mathbb{E}\Big[\mathbf{I}\{\pi_f(x) \ne \pi_{f^*}(x)\}(f^*(x, \pi_{f^*}(x)) - f(x, \pi_{f^*}(x)) + f(x, \pi_{f^*}(x)) - f(x, \pi_f(x)) \\
&\quad + f(x, \pi_f(x)) - f^*(x, \pi_f(x)))\Big] \\
\le&\mathbb{E}[\mathbf{I}\{\pi_f(x) \ne \pi_{f^*}(x)\}(|f^*(x, \pi_{f^*}(x)) - f(x, \pi_{f^*}(x))| + |f^*(x, \pi_f(x)) - f(x, \pi_f(x))|)].
\end{aligned}
$$

We now consider the term $(Reg_t(\pi_f))^2$ as following.

$$
\begin{aligned}
&(Reg_t(\pi_f))^2 \\
\le&(\mathbb{E}[\mathbf{I}\{\pi_f(x) \ne \pi_{f^*}(x)\}(|f^*(x, \pi_{f^*}(x)) - f(x, \pi_{f^*}(x))| + |f^*(x, \pi_f(x)) - f(x, \pi_f(x))|)])^2 \\
\le&\mathbb{E}\left[\left(\frac{\mathbf{I}\{|S_t| > 1\}}{p_{t-1}(\pi_f(x)|x, S_t)} + \frac{\mathbf{I}\{|S_t| > 1\}}{p_{t-1}(\pi_{f^*}(x)|x, S_t)}\right)\mathbb{E}_{x,a\sim p_{t-1}(\cdot|x,S_t)}(f^*(x, a) - f(x, a))^2\right] \\
\le&\mathbb{E}\left[\left(\frac{\mathbf{I}\{|S_t| > 1\}}{p_{t-1}(\pi_f(x)|x, S_t)} + \frac{\mathbf{I}\{|S_t| > 1\}}{p_{t-1}(\pi_{f^*}(x)|x, S_t)}\right)\right]\frac{(2\beta_m + C_\delta)}{n_m/2} \\
\le&\mathbb{E}\left[\left(\frac{\mathbf{I}\{|S_t| > 1\}}{p_{t-1}(\pi_f(x)|x, S_t)} + \frac{\mathbf{I}\{|S_t| > 1\}}{p_{t-1}(\pi_{f^*}(x)|x, S_t)}\right)\right]\frac{2M\eta\lambda_m^2}{\rho_m^2}.
\end{aligned}
$$

From Corollary B.1 we know

$$
\begin{aligned}
&\mathbb{E}\left[\frac{\mathbf{I}\{|S_t| > 1\}}{p_{t-1}(\pi_f(x)|x, S_t)}\right] \\
\le&q_{m-1}\mathbb{E}_G[\alpha(G)] + \sqrt{\mathbb{E}[\alpha(G)]}\rho_{m-1}\widehat{Reg}_{\tau_{m-1}}(\pi_f) \\
\le&q_{m-1}\mathbb{E}_G[\alpha(G)] + 2\sqrt{\mathbb{E}[\alpha(G)]}\rho_{m-1}Reg_{\tau_{m-1}}(\pi_f) + c_0\hat{w}_{m-1}\mathbb{E}[\alpha(G)] \\
\le&\frac{3}{2}\hat{w}_{m-1}\mathbb{E}_G[\alpha(G)] + 2\sqrt{\mathbb{E}[\alpha(G)]}\rho_{m-1}Reg_{\tau_{m-1}}(\pi_f) + c_0\hat{w}_{m-1}\mathbb{E}[\alpha(G)].
\end{aligned}
$$

and

$$
\begin{aligned}
&\mathbb{E}\left[\frac{\mathbf{I}\{|S_t| > 1\}}{p_{t-1}(\pi_{f^*}(x)|x, S_t)}\right] \\
\le&q_{m-1}\mathbb{E}_G[\alpha(G)] + \sqrt{\mathbb{E}[\alpha(G)]}\rho_{m-1}\widehat{Reg}_{\tau_{m-1}}(\pi_{f^*}) \\
\le&q_{m-1}\mathbb{E}_G[\alpha(G)] + 2\sqrt{\mathbb{E}[\alpha(G)]}\rho_{m-1}Reg_{\tau_{m-1}}(\pi_{f^*}) + c_0\hat{w}_{m-1}\mathbb{E}[\alpha(G)] \\
\le&\frac{3}{2}\hat{w}_{m-1}\mathbb{E}_G[\alpha(G)] + 72\hat{w}_{m-1}\mathbb{E}_G[\alpha(G)].
\end{aligned}
$$

Plugging the above two inequalities yields

$$(Reg_t(\pi_f))^2 \le (2\sqrt{\mathbb{E}_G[\alpha(G)]}\rho_{m-1}Reg_t(\pi_f) + (2c_0 + 3)\hat{w}_{m-1}\mathbb{E}_G[\alpha(G)])\frac{2M\eta\hat{w}_m^2}{\hat{w}_{m-1}\rho_m^2}.$$

which implies

$$(Reg_t(\pi_f))^2 \le (2\sqrt{\mathbb{E}_G[\alpha(G)]}\frac{\rho_m\hat{w}_{m-1}}{\hat{w}_m}Reg_t(\pi_f) + (2c_0 + 3)\hat{w}_{m-1}\mathbb{E}_G[\alpha(G)])\frac{2M\eta\hat{w}_m^2}{\hat{w}_{m-1}\rho_m^2}$$

according to Proposition B.1.

Solving the inequality for $Reg_t(\pi_f)$ shows that

$$Reg_t(\pi_f) \leq c_1 \hat{w}_m \log T \sqrt{\mathbb{E}_G[\alpha(G)]}/\rho_m,$$

where $c_1 = 6$.

$\square$

At this point, we can bound the regret within each epoch using the above, which gives a bound in terms of the empirical disagreement probability $\hat{w}_m$. To proceed, we relate this quantity to the policy disagreement coefficient. Our proof can directly follow from that in Foster et al. (2020) by replacing $A$ with $\mathbb{E}_G[\alpha(G)]$ and $A/\gamma_m$ with $\sqrt{\mathbb{E}_G[\alpha(G)]}/\rho_m$. Our regret analysis builds on the framework in Simchi-Levi & Xu (2021). Hence, we can directly write down the following lemma.

**Lemma B.8** *Assume that $\Gamma$ holds. For any fix $\epsilon > 0$ and every $m \in [M]$, we have*

$$\sum_{\pi \in \Psi} Q_t(\pi) Reg_t(\pi) \leq \max\left\{\epsilon, \theta^{csc}(\mathcal{F}, \epsilon) \frac{c_2 \mathbb{E}[\alpha(G)] \log(2\delta^{-1}T^2|\mathcal{F}|)}{n_{m-1}}\right\} + \frac{256 \log(4M/\delta)}{n_{m-1}},$$

*where $\theta^{csc}(\mathcal{F}, \epsilon) = \sup_{\epsilon \geq \epsilon_0} \frac{1}{\epsilon} \mathbb{P}_{\mathcal{X}}(x \in \mathcal{X} : \exists f \in \mathcal{F}_\epsilon^{csc}$ such that $\pi_f(x) \neq \pi_{f^*}(x))$ and $\mathcal{F}_\epsilon^{csc} = \{f \in \mathcal{F}|\mathcal{R}(\pi_{f^*}) - \mathcal{R}(\pi_f) \leq \epsilon\}$.*

**Corollary B.2** *Assume that $\Gamma$ and Assumption 3.1 holds. For any fix $\epsilon > 0$ and every $m \in [M]$, we have*

$$\sum_{\pi \in \Psi} Q_t(\pi) Reg_t(\pi) \leq \max\left\{\epsilon\Delta, \frac{c_2 \theta^{pol}(\mathcal{F}, \epsilon) \mathbb{E}[\alpha(G)] \log(2\delta^{-1}T^2|\mathcal{F}|)}{\Delta n_{m-1}}\right\} + \frac{256 \log(4M/\delta)}{n_{m-1}}.$$

*Proof.* We replace $\epsilon\Delta$ with $\epsilon$ in Lemma B.8. Since

$$\mathcal{F}_{\epsilon\Delta}^{csc} = \{f \in \mathcal{F}|\mathcal{R}(\pi_{f^*}) - \mathcal{R}(\pi_f) \leq \epsilon\Delta\},$$

we have

$$\mathbb{P}_{\mathcal{X}}(x \in \mathcal{X} : \exists f \in \mathcal{F}_\epsilon \text{ such that } \pi_f(x) \neq \pi_{f^*}(x))\Delta \leq \mathcal{R}(\pi_{f^*}) - \mathcal{R}(\pi_f) \leq \epsilon\Delta.$$

We conclude that

$$\mathbb{P}_{\mathcal{X}}(x \in \mathcal{X} : \exists f \in \mathcal{F}_\epsilon \text{ such that } \pi_f(x) \neq \pi_{f^*}(x)) \leq \epsilon$$

and thus

$$\theta^{csc}(\mathcal{F}, \epsilon\Delta) \leq \sup_{\epsilon \geq \epsilon'} \frac{\mathbb{P}_{\mathcal{X}}(x \in \mathcal{X} : \exists f \in \mathcal{F}_{\epsilon'} \text{ such that } \pi_f(x) \neq \pi_{f^*}(x))}{\epsilon'\Delta} = \theta^{pol}(\mathcal{F}, \epsilon)/\Delta.$$

$\square$

Now it is time to prove the gap-dependent upper bound in Theorem 3.3.

*Proof.*

$$\mathbb{E}[Reg(T)] = \sum_{t=1}^{T} \sum_{\pi \in \Psi} Q_{m(t)}(\pi) Reg_t(\pi)$$

$$\leq \sum_{m=1}^{M} \sum_{t=\tau_{m-1}+1}^{\tau_m} \sum_{\pi \in \Psi} Q_t(\pi) Reg_t(\pi)$$

$$\leq \sum_{m=1}^{M} n_m \left(\max\{\epsilon\Delta, \frac{c_2 \theta^{pol}(\mathcal{F}, \epsilon) \mathbb{E}[\alpha(G)] \log(2\delta^{-1}T^2)}{\Delta n_{m-1}}\} + \frac{256 \log(4M/\delta)}{n_{m-1}}\right)$$

$$\leq \sum_{m=1}^{M} \left(\max\{\epsilon\Delta n_m, \frac{2c_2 \theta^{pol}(\mathcal{F}, \epsilon) \mathbb{E}[\alpha(G)] \log(2\delta^{-1}T^2)}{\Delta}\} + 512 \log(4M/\delta)\right)$$

$$\leq \max\left\{\epsilon\Delta T, \frac{2c_2 \theta^{pol}(\mathcal{F}, \epsilon) \mathbb{E}[\alpha(G)] \log(2\delta^{-1}T^2) \log T}{\Delta}\right\} + 512 \log T \log(4 \log T/\delta)$$

$\square$

Again, for the second option, we can directly replace $\mathbb{E}_G[\alpha(G)]$ with $\mathbb{E}_G[\delta_f(G)]$.

### B.5  LOWER BOUND

To establish tour lower bound, we synergistically combine the methodologies employed to prove lower bounds in contextual bandits as described in (Foster et al., 2020), with the techniques utilized in proving lower bounds for multi-armed bandit algorithms incorporating graph feedback, as presented in (Buccapatnam et al., 2017). We introduce hyperparameters $\Delta$ and $\epsilon$ for our construction, which will be determined later. For simplicity, we assume that the action set $\mathcal{A}$ can be rewritten as $[|\mathcal{A}|]$.

**Contexts.** We define $d := \lceil \log(|\mathcal{F}|) \rceil$ and $k = \lfloor \epsilon^{-1} \rfloor$. The context domain, denoted as $\mathcal{X}$, is constructed from $d$ disjoint partitions labeled as $\mathcal{X}^{(1)}, \cdots, \mathcal{X}^{(d)}$. Each partition $\mathcal{X}^{(i)}$ comprises the set $\{x^{(i,0)}, x^{(i,1)}, \cdots, x^{(i,k)}\}$ for $i \in [d]$. The total size of different contexts is $d(k+1)$. By taking the union of these partitions, we obtain the complete context domain $\mathcal{X}$.

To define the context distribution $D$, we specify the probabilities assigned to each context in $\mathcal{X}$. Let $D^i$ denote the distribution over $\mathcal{X}^{(i)}$. In $D^i$, each context $x^{(i,j)}$ for $j \in [k]$ is assigned a probability of $\epsilon$, while the context $x^{(i,0)}$ has a probability of $1 - k\epsilon \geq 0$. To obtain the overall context distribution $D$, we average $D^i$ over all $d$ partitions: $D = \frac{1}{d} \sum_{i=1}^{d} D^i$.

**Function space.** We now choose a regression function class $\mathcal{F}$. For each subset $\mathcal{X}^{(i)}$, we define a corresponding class of regression functions $\mathcal{F}^{(i)}$ in the following manner. Initially, we set $f^{(i,0)}(x^{(i,j)}, \cdot) = (1/2 + \Delta, 1/2, \cdots, 1/2)$ for all $j$. Next, for each $b \neq a_1$ and $l \in [k]$, we specify:

$$f^{(i,l,b)}(x^{(i,j)}, \cdot) = \left\{ \begin{array}{l} \frac{1}{2}\mathbf{1} + \Delta e_1, j \neq l \\ \frac{1}{2}\mathbf{1} + \Delta e_b, j = l \end{array} \right.$$

Here, $\mathbf{1}$ denotes all-one vectors and $e_b$ is a basis vector where the b-coordinate is equal to 1. To obtain the function class $\mathcal{F}$, we stitch together $\mathcal{F}^{(1)}, \cdots, \mathcal{F}^{(d)}$ over their respective subsets of the domain.

The function class $\mathcal{F}$ induces a policy space denoted as $\Pi$. This policy space satisfies $\pi^{(i,l,b)}(x^{(i,0)}) = 1$ and

$$\pi^{(i,l,b)} = \left\{ \begin{array}{ll} 1, & \text{if } j \neq l, \\ b, & \text{if } j = l. \end{array} \right.$$

This class consists of all policies that deviate from $a_1$ on a subset of contexts of size at most $d$, with the condition that this subset intersects with each $\mathcal{X}^{(i)}$ at most once.

**Regret decomposition.** Let $\mathbb{P}^{(i,l,b)}$ denote the reward distribution given by $r(a) \sim$ Bernoulli($f^{(i,l,b)}(x,a)$) conditioned on $x$ for each $x \in \mathcal{X}^{(i)}$. We define $p_t(x,a) = \mathbb{P}(a_t = a | \mathcal{H}t - 1, x_t = x)$ and $\bar{p}(x,a) = \frac{1}{T} \sum_{t=1}^{T} p_t(x,a)$. For any sequence $\nu = \nu_1, \cdots, \nu_d$, where $\nu_i = (v_i, b_i)$ with $v_i \in [k]$ and $b_i \in \{2, \cdots, |\mathcal{A}|\}$, we let $\mathbb{P}_\nu$ denote the distribution of $\mathcal{H}_T$ when the reward distribution for $\mathcal{X}^{(i)}$ is given by $\mathbb{P}^{(i,v_i,b_i)}$. We sample the problem instance $\nu$ from a distribution defined as follows: for each $i$, set $v_i = 0$ with probability $1/2$. Otherwise, select $v_i$ uniformly from $[k]$, and select $b_i$ uniformly from $\{2, \cdots, |\mathcal{A}|\}$. According to Foster et al. (2020), we have the inequality

$$\mathbb{E}[\mathbb{E}_\nu[Reg(T)]] \geq \frac{\Delta T}{4kd(|\mathcal{A}| - 1)} \sum_{i=1}^{d} \sum_{l=1}^{k} \sum_{b \neq a_1} \mathbb{E}_{\mathbb{P}^{(i,l,b)}} \|\bar{p} - \pi^{(i,l,b)}\|_{L_1(D^{(i)})}.$$

Let $\mathcal{I} \subset [d]$ denote the set of indices $i$ for which

$$\frac{1}{k(|\mathcal{A}| - 1)} \sum_{l=1}^{k} \sum_{b \neq a_1} \mathbb{E}_{\mathbb{P}^{(i,l,b)}} \|\bar{p} - \pi^{(i,l,b)}\|_{L_1(D^{(i)})} \leq \epsilon/32.$$

If $|\mathcal{I}| \leq d/2$, then

$$\mathbb{E}[\mathbb{E}_\nu[Reg(T)]] \geq \epsilon\Delta T/32.$$

For the other case, we have $|\mathcal{I}| \geq d/2$. In this case, such condition satisfies the requirement of the Fano method with reverse KL-divergence in (Raginsky & Rakhlin, 2011). It implies that for $\mathbb{P}^{(i,0,0)}$ we have

$$\ln 2 \leq \frac{1}{k} \sum_{l=1}^{k} KL(\mathbb{P}^{(i,0,0)} \| \mathbb{P}^{(i,l,b)}), \forall b = 2, 3, \cdots, |\mathcal{A}|.$$

Note that

$$
\begin{aligned}
& KL(\mathbb{P}^{(i,0,0)}\|\mathbb{P}^{(i,l,b)}) \\
= \ & KL(Bernoulli(1/2)\|Bernoulli(1/2+2\Delta))\mathbb{E}_{\mathbb{P}^{(i,0,0)}}[O(x^{(i,l)},b)] \\
\leq \ & 4\Delta^2 \mathbb{E}_{\mathbb{P}^{(i,0,0)}}[O(x^{(i,l)},b)],
\end{aligned}
$$

where $O(x^{(i,l)},b)$ is equal to the number of observing the rewards of the arm $b$ when the context $x^{(i,l)}$ occurs. Hence, we have

$$
\mathbb{E}_{\mathbb{P}^{(i,0,0)}}[O(\mathcal{X}^{(i)} - \{x^{(i,0)}\},b)] \geq \frac{k\log 2}{4\Delta^2}.
$$

Denote $N(\mathcal{X}^{(i)} - \{x^{(i,0)}\},b)$ the number of pulling the arm $b$ when the contexts in $\mathcal{X}^{(i)} - \{x^{(i,0)}\}$ occur. From the graph structure we know that

$$
\sum_{u\in\mathcal{N}_b} g_{u,b}\mathbb{E}_{\mathbb{P}^{(i,0,0)}}[N(x_t \in \mathcal{X}^{(i)} - \{x^{(i,0)}\},u)] \geq \mathbb{E}_{\mathbb{P}^{(i,0,0)}}[O(x_t \in \mathcal{X}^{(i)} - \{x^{(i,0)}\},b)] \geq \frac{k\log 2}{4\Delta^2}
$$

These lead to the constraints of the linear programming (2). Moreover, we have

$$
\mathbb{E}[\mathbb{E}_\nu[Reg(T)]] \geq \Delta\mathbb{E}\left[\mathbb{E}_\nu[\sum_{t=1}^{T}\sum_{i=1}^{d}\mathbf{I}\left\{x_t \in \mathcal{X}^{(i)} - \{x^{(i,0)}\}, a_t \neq a_1, v_i = 0\right\}]\right]
$$

$$
\geq \frac{\Delta}{2}\sum_{i=1}^{d}\sum_{t=1}^{T}\mathbb{E}_{\mathbb{P}^{(i,0,0)}}[\mathbf{I}\left\{x_t \in \mathcal{X}^{(i)} - \{x^{(i,0)}\}, a_t \neq a_1\right\}]
$$

$$
= \frac{\Delta}{2}\sum_{i=1}^{d}\sum_{b\neq a_1}\mathbb{E}_{\mathbb{P}^{(i,0,0)}}[N(\mathcal{X}^{(i)} - \{x^{(i,0)}\},b)].
$$

The term $\sum_{b=1}^{|\mathcal{A}|}\mathbb{E}_{\mathbb{P}^{(i,0,0)}}[N(\mathcal{X}^{(i)} - \{x^{(i,0)}\},b)]$ is the object of the linear programming, so

$$
\sum_{b\neq a_1}\mathbb{E}_{\mathbb{P}^{(i,0,0)}}[N(\mathcal{X}^{(i)} - \{x^{(i,0)}\},b)] \geq \frac{k\delta_f(G)\ln 2}{4\Delta^2} = \frac{k\delta_f(G)}{8\Delta^2}
$$

if $\mathbb{E}_{\mathbb{P}^{(i,0,0)}}[N(\mathcal{X}^{(i)} - \{x^{(i,0)}\},1)] \geq |\mathcal{A}|$.

Combing all above inequalities yield

$$
\mathbb{E}[\mathbb{E}_\nu[Reg(T)]] \geq \frac{k\delta_f(G)}{16\Delta} \geq \frac{\delta_f(G)}{32\epsilon\Delta},
$$

where we can select a value $\epsilon$ such that $\frac{1}{\epsilon} \geq \frac{1}{2}\lfloor\frac{1}{\epsilon}\rfloor$. Therefore we have

$$
\mathbb{E}[\mathbb{E}_\nu[Reg(T)]] \geq \min\{\frac{\delta_f(G)}{32\epsilon\Delta}, \frac{\epsilon\Delta T}{32}\}.
$$

Now we choose $\epsilon\Delta = \frac{\delta_f(G)}{T}$ and obtain

$$
\mathbb{E}[\mathbb{E}_\nu[Reg(T)]] \geq \sqrt{\delta_f(G)T}/32.
$$

Since the expectation is larger than $\sqrt{\delta_f(G)T}/32$, There must exist an instance such that the regret on this instance is larger than $\sqrt{\delta_f(G)T}/32$.

**Time-varying graphs.** Now we consider the time-varying graphs. We divide the time horizon according to the graph $G$. We know that there exists an instance such that

$$
\mathbb{E}[Reg(T_G)] \geq \sqrt{\delta_f(G)T_G}/32,
$$

where $T_G$ is the number of times that the graph $G$ occurs. Therefore,

$$\mathbb{E}[Reg(T)] = \mathbb{E}_G[\sum_{G \in \mathcal{G}} \mathbb{E}_G[Reg(T_G)|G]]$$

$$\geq \frac{1}{32}\mathbb{E}[\sum_{G \in \mathcal{G}} \sqrt{\delta_f(G)\mathbb{E}[T_G|G]}]$$

$$\geq \frac{1}{32}\sqrt{\mathbb{E}[\sum_{G \in \mathcal{G}} \delta_f(G)\mathbb{E}[T_G|G]]}$$

$$\geq \frac{1}{32}\sqrt{\mathbb{E}[\sum_{G \in \mathcal{G}} \delta_f(G)T\mathbb{P}(G)]}$$

$$\geq \frac{1}{32}\sqrt{\mathbb{E}_G[\delta_f(G)]T}.$$

We complete the proof.

## C  ADDITIONAL ALGORITHMS

---

**Algorithm 2** ConstructExplorationSet - Option 1

---

**Input:** the adjacency matrix $G_t$, the adaptive set $\mathcal{A}(x_t; \mathcal{F}_m)$, gaps $\Delta_{a,t}$ for $a \in \mathcal{A}(x_t; \mathcal{F}_m)$
 1: Sort gaps in the ascending order
 2: Initialize auxiliary set $B_t$ and the exploration set $S_t$ to be empty
 3: **for** each arm $a$ in $\mathcal{A}(x_t; \mathcal{F}_m)$ **do**
 4:     **if** the arm $a$ is not in $B_t$ **then**
 5:         Put the arm $a$ in $S_t$
 6:         Update $B_t = B_t \cup \mathcal{N}_a(G_t)$
**Output:** An exploration set $S_t$

---

**Algorithm 3** ConstructExplorationSet - Option 2

---

**Input:** the adjacency matrix $G_t$, the adaptive set $\mathcal{A}(x_t; \mathcal{F}_m)$, the empirical best arm $\hat{a}_t$
 1: Solve the (2) to obtain the optimal solution $z_a^*$ for each $a \in \mathcal{A}$
 2: Initialize the exploration set $S_t$ to be empty
 3: **for** each arm $a$ in $\mathcal{A}$ **do**
 4:     **if** Bernoulli($z_a^*$) == 1 **then**
 5:         Put the arm $a$ in $S_t$
 6: **for** each arm $a$ in $S_t$ **do**
 7:     **if** $\mathcal{N}_a(G_t) \cap \mathcal{A}(x_t; \mathcal{F}_m) = \emptyset$ **then**
 8:         Remove the arm $a$ in $S_t$
 9:     **if** $|\mathcal{N}_a(G_t) \cap \mathcal{A}(x_t; \mathcal{F}_m)| = 1$ **then**
10:         Let the arm in $\mathcal{N}_a(G_t) \cap \mathcal{A}(x_t; \mathcal{F}_m)$ be $\tilde{a}$
11:         Remove the arm $a$ in $S_t$ and add the arm $\tilde{a}$ in $S_t$
12: Add the empirical best arm $\hat{a}_t$ in $S_t$
**Output:** An exploration set $S_t$

---

**Algorithm 4** A random graph generator

---

**Input:** The number of nodes $K$, a dense factor $\eta$
 1: Initialization: a $K \times K$ identity matrix $G$, a counter $t$
 2: **repeat**
 3:     Uniformly sample two nodes $u, v$ in $[K]$
 4:     Add the edge $(u, v)$ and $(v, u)$, i.e., $G[u][v] = G[v][u] = 1$
 5:     $t = t + 1$
 6: **until** $t \geq \eta \times K^2$
**Output:** An adjacency matrix $G$

---

## D  CONCLUSION

In this paper, we have introduced a framework for incorporating side-observations into contextual bandits with a general reward function space. We have derived instance-independent upper and lower bounds on the regret and proposed a near-optimal algorithm that matches these lower bounds up to logarithmic terms and constants. However, there are several avenues for future research and extension of our work.

Firstly, it would be valuable to explore the possibility of capturing the gap-dependent upper bound in a more precise manner than what is presented in Theorem 3.3. Assuming the gap condition:

$$f^*(x, \pi_{f^*}(x)) - f^*(x, a) \geq \Delta_a, \quad \forall x \in \mathcal{X},$$

we still lack a method to establish a gap-dependent upper bound specific to each arm, similar to the case in MAB. Obtaining such gap-dependent upper bounds would allow us to more accurately

---

**Algorithm 5** An Adaptive Contextual Bandit algorithm with Graph feedback (AdaCB.G)

---

**Input:** time horizon $T$, confidence parameter $\delta$, tuning parameters $\eta$

1: Set epoch schedule $\{\tau_m = 2^m, \forall m \in \mathbb{N}\}$ and the sample splitting schedule $t_m = \frac{\tau_m + \tau_{m-1}}{2}$

2: **for** epoch $m = 1, 2, \cdots, \lceil \log_2 T \rceil$ **do**

3:      Compute the confidence radius $\beta_m = 16(\log T - m + 1)\log(2|\mathcal{F}||\mathcal{A}|^2 T^2/\delta)$

4:      Compute the smoothing parameter $\mu_m = 64\log(4\delta^{-1}\log T)/(\tau_m - \tau_{m-1})$

5:      Compute the function

$$\hat{f}_m = \arg\min_{f \in \mathcal{F}} \sum_{n=1}^{\tau_{m-1}} \sum_{a \in \mathcal{N}_{a_n}(G_n)} (f(x_n, a) - y_{n,a})^2$$

     via the **Offline Least Square Oracle**

6:      Compute

$$\mathcal{F}_m = \left\{ f \in \mathcal{F} \mid \sum_{n=1}^{t_{m-1}} \sum_{a \in \mathcal{N}_{a_n}(G_n)} (f(x_n, a) - y_{n,a})^2 \leq \min_{\tilde{f} \in \mathcal{F}} \sum_{n=1}^{t_{m-1}} \sum_{a \in \mathcal{N}_{a_n}(G_n)} (\tilde{f}(x_n, a) - y_{n,a})^2 + \beta_m \right\}$$

7:      Compute the instance-dependent scale factor

$$\lambda_m = \frac{\mathbb{E}_{x \sim \mathcal{D}_m}[\mathbf{I}\{\mathcal{A}(x; \mathcal{F}_m) > 1\}] + \mu_m}{\sqrt{\mathbb{E}_{x \sim \mathcal{D}_{m-1}}[\mathbf{I}\{\mathcal{A}(x; \mathcal{F}_{m-1}) > 1\}] + \mu_{m-1}}},$$

     where $\mathcal{D}_m \sim unif(x_{t_{m-1}+1, \cdots, x_{\tau_{m-1}}})$ (for the first epoch, $\lambda_1 = 1$)

8:      **for** round $t = \tau_{m-1} + 1, \cdots, \tau_m$ **do**

9:          Observe the context $x_t$ and the graph $G_t$

10:         Compute the best arm candidate set $\mathcal{A}(x_t; \mathcal{F}_m)$

11:         **if** $|\mathcal{A}(x_t; \mathcal{F}_m)| == 1$ **then**

12:             Let the exploration set be $S_t = \{\hat{a}_t\}$, where $\hat{a}_t = \max_{a \in \mathcal{A}} \hat{f}_m(x_t, a)$

13:         **else**

14:             Call the subroutine **ConstructExplorationSet** to find the exploration set $S_t$

15:             **if** $|S_t| \geq |\mathcal{A}(x_t; \mathcal{F}_m)|$ **then**

16:                 Let the exploration set $S_t$ be $\mathcal{A}(x_t; \mathcal{F}_m)$

17:         Compute $\gamma_t = \lambda_m \sqrt{\frac{\eta|S_t|\tau_{m-1}}{2\log(2\delta^{-1}|\mathcal{F}|T^2)}}$ (for the first epoch, $\gamma_t = 0$)

18:         Compute the following probabilities

$$p_t(a) = \begin{cases} \frac{1}{|S_t| + \gamma_t(\hat{f}_m(x_t, \hat{a}_t) - \hat{f}_m(x_t, a))}, & \text{for all } a \in S_t - \{\hat{a}_t\} \\ 0, \text{ for all } a \in \mathcal{A} - S_t, \\ 1 - \sum_{a \neq \hat{a}_t} p_t(a), \text{ for } a = \hat{a}_t, \end{cases}$$

     where $\hat{a}_t = \max_{a \in \mathcal{A}} \hat{f}_m(x_t, a)$

19:          Sample $a_t \sim p_t(\cdot)$ and take the action $a_t$

20:          Observe a feedback graph $\{(a, y_{t,a}) | a \in \mathcal{N}_{a_t}(G_t)\}$ from $G_t$

---

balance the trade-off between the number of arms to explore and their corresponding gaps, as we discuss in the first option and the second option.

Then, we aim to develop a best-of-both-worlds algorithm that can automatically adapt and perform well in both stochastic and adversarial settings. This will enhance the practical applicability of our framework and allow it to address a broader range of real-world problems. By designing an algorithm that can dynamically adjust its behavior based on the environmental characteristics, people may achieve superior performance across different scenarios.

Furthermore, we recognize the importance of studying uninformed graph feedback problems. While our current framework focuses on the informed graph feedback setting, where the entire feedback graph is known prior to each decision, many practical problems involve uninformed graph feedback, where the graph is unknown at decision time. Investigating strategies and algorithms that can handle this scenario effectively is an important direction for future research.

Additionally, we aim to derive gap-dependent upper bounds for more general types of feedback graphs, such as weakly observable graphs. Our current approach may face limitations in constructing an effective exploration set in these cases, and new techniques and strategies need to be developed to overcome these challenges. By extending our framework to handle diverse types of feedback graphs, we can enhance its versatility and address a wider range of real-world applications.