# OpenReview forum: "Improved Regret Bounds in Stochastic Contextual Bandits with Graph Feedback"
_ICLR.cc/2024/Conference — ICLR 2024 Conference Withdrawn Submission_

### Official Review · Reviewer_Z4bM · 2023-10-21

**Soundness:** 1 poor
**Presentation:** 2 fair
**Contribution:** 1 poor
**Rating:** 1
**Confidence:** 4

**Summary:**

This paper considers the problem of (stochastic) contextual bandits under feedback graphs, which generalizes the classic contextual bandit problem. Specifically, the authors consider the realizable setting [Foster et al., 2018, Foster et al., 2020] where the true expected loss function $f^\star$ lies in a function class $\mathcal{F}$, which is known to the learner. The authors proposed an algorithm based on FALCON [Simchi-Levi & Xu 2022] but with a different choice of exploration set $S_t$ and claimed to achieve $O(\sqrt{\delta T})$ regret where $\delta$ is the averaged expected domination number of feedback graph. This is achieved by selecting $S_t$ to be the dominating set of the graph. The author also shows that with an adaptive tuning technique proposed in [Foster et al., 2020], their proposed algorithm achieves the gap dependent bound $\tilde{O}(\delta/\Delta)$. Moreover, the authors also prove that the problem-independent lower bound is $\Omega(\sqrt{\delta T})$.

**Strengths:**

- The problem considered in this paper is important and the motivation is clearly stated.

**Weaknesses:**

- This paper is not written clearly and does not provide clear proofs for the claimed theorems. Specifically, I do not find proofs about the $O(\sqrt{\delta T})$ regret upper bound in the appendix. In fact, this $O(\sqrt{\delta T})$ result just should not hold since it breaks the lower bound proven in [Alon et al., 2015, 2017] even in the non-contextual setting. Consider the star graph (or K-tree graph in the context of this paper). If the center node has 1 loss and the remaining nodes have $Ber(1/2)$ loss except for one node having $Ber(1/2-\epsilon)$ loss, we can not do better than $\min(\sqrt{KT}, T^{2/3})$ regret bound but what claimed in this paper is a $O(\sqrt{T})$ regret bound. Although the authors do not show the exact analysis in the appendix for the $O(\sqrt{\delta T})$ result, technically, the error is that their Lemma B.2, Lemma B.3 both consider the policy on $\Psi(S_t)$ but the regret benchmark may not be within this set, making the analysis break.

- For the upper bound result achieving $O(\sqrt{\alpha T})$ regret, option 1 for ConstructExplorationSet is actually also not new and is shown in Proposition 4 of [Zhang et al., 2023] for self-aware undirected graphs. Therefore, I feel that the result with respect to the independence number is also not hard to obtain based on FALCON and this exploration set construction.

**Questions:**

Can authors provide detailed proofs for the results claimed in the paper, especially for the upper bound results that is related to the dominating number?

**Details Of Ethics Concerns:**

None.

---

### Official Review · Reviewer_7kWt · 2023-10-30

**Soundness:** 3 good
**Presentation:** 2 fair
**Contribution:** 2 fair
**Rating:** 6
**Confidence:** 4

**Summary:**

In this paper, the authors consider a contextual bandits framework with a general reward function space and side-observations given to the learner in form of (informed) feedback graphs. The authors propose a new algorithm that achieves an improved upper-bound (instance-independent) regret upper-bound than previously known algorithms.  The authors proceed to prove a lower-bound to show the near-optimality of this algorithm (ignoring logarithmic terms and constants). Finally, some modifications are introduced in order to derive a gap-dependent upper-bound. Several experiments are conducted to highlight the improvements of proposed algorithms (and further analyses are realized to show the limitations of previously known algorithms).

**Strengths:**

As I understand, the paper poses a quite interesting question on the extension of contextual bandit framework with general reward function space (and the associated inverse gap weighting technique) into scenarios with graphical feedbacks. In general, the writing is quite good (although the structure is less so). The authors attempt to provide both concrete proofs and intuition explanation, this is applaudable.

Moreover, the authors have done a quite comprehensive comparison with three previous works that facilitates the review tasks.  The authors also keep both options in building the exploration set (which seems to be the key ingredient of the proposed algorithm) and analyse the involved exploration-exploitation trade-off (on top of the traditional trade-off of bandit); which is rather quite interesting although this make the presentation more cumbersome (see below).

**Weaknesses:**

- The paper is not very well structured; it is not easy to understand the flow of the paper. In particular, although Table 1 and 2 capture quite well the main contributions, the involved notation are not introduced there but scattering across the paper (hence, readers need to revise these tables after finishing the paper to understand the notation). Similarly, while it is good to have a comparison with previous works in Section 1.2, it is hard to understand as no proper description of the contribution is introduced yet (for example, the beginning of page 4 discusses “first and second options” that has never been mentioned before). A simplified version of main theorem before this section, for example, could facilitate the comprehension here.
- Another major critic is that it does not seem necessary to present ConstructExplorationSet with two options (or at least it lacks a major justification for the necessity of this). As I understand from page 6, “Option 2” is recommended to be used with an integration of empirical best arms from Option 1. Moreover, from Table 1, the major improvement (switching from bounds with independence numbers to bounds with dominating numbers) comes from Option 2. The experiments also show the speriority of this Option 2. I do not see why the authors cannot simply combine these two so-call options into one procedure.
- The notion delta_f of fractional dominating number is important to this paper, but it is never defined properly. As mentioned above, the ability to obtain a method having bounds with this delta_f instead of alpha is a major point; however, the proof is written only for the alpha case (so-called option 1) and the detailed proof for delta_f is omitted in appendices. Only a small explanation is presented in page 6 that is not sufficient.
- The idea of the main algorithm is quite natural and obvious (applying IGW to a well-selected exploration set). Can the authors highlight further any novelty of this algorithm or the contribution comes more in the proof aspects?
- Another point that should be mentioned is that the main result is of the high-probability bound flavour that differs from most of previous works that are directed compared.

**Questions:**

- Technically, in Tables 1 and 2, results of Wang et al.2021 can be presented with upper-bound of alpha(G_t) and hence, only require to know this upper-bound instead of the real independence number. This mitigate the "critics" in page 4.

- Do the authors choose to run experiements only in comparison with FALCON and not the one of Zhang et al. 2023 because the latter is instable?

- Why do the authors prefer high-probability bound? Is it posible/easy to derive an expected regret result from Theorem 3.1?

- The presented framework uses undirected feedback graph, can this be extended with directed ones? (note that Zhang et al. consider directed graph).

- Can we have a definition of the set S_t?

**Details Of Ethics Concerns:**

No ethical issues with this paper to signify.

---

> ### Author Response · Authors · 2023-11-22
>
> Q: The paper is not very well structured
>
> A: Thanks for suggestions. To help readers understand our work better, we add the name of graph parameters in the caption of tables.
> We also show the order sequence of this graph parameters to highlight our contributions.
> As for ``a simplified version of main theorem'', we will try to put it in our work if there exists additional space.
> After all, the limited page requirement of ICLR forces us to put formal descriptions in the following sections.
>
>
> Q: it does not seem necessary to present ConstructExplorationSet with two options
>
> A: Our main contribution stems from the second option.
> However, the first option is not always useless, as in certain types of graphs like perfect graphs,
> it can outperform the second option.
> The two options also capture the trade-off of the exploration between the gaps and number of actions.
> Additionally, the first option can be served as a rough realization of the method in [1] as their regret order scales with the independence number.
>
>
>
> Q: the contribution comes more in the proof aspects
>
> A: It requires non-trivial modifications of IGW technique.
> The time-varying hyperparameter $\gamma_t$ and the adaptive set $S_t$ are the crucial difference of original works.
> The original works focus on the upper bound so they only need an upper bound $|\mathcal{A}|$.
> In order to capture the influence of time-varying action sets,
> we have to bound the size of action sets in a more refined way.
> It requires us to define the policy space and prove lemmas in a more refined way.
>
> Here is the high-level intuition of our proof.
> For any policy $\pi \in \Psi(S_t)$, the regret of $\pi$ with respect to the greedy policy can be controlled by our lemma B.1.
> Secondly, since the sample size will increase as the learning processes, the regret of the greedy policy with respect to the optimal policy will become smaller,
> as shown in lemma B.2.
> Due to these two lemmas,
> the policy $\pi \in \Psi(S_t)$ can be controlled.
>
>
>
> Q: the high-probability bound to the expected bound
>
> A: It can be converted to expected bounds by setting $\delta=1/T$.
>
>
>
> Q: Technically, in Tables 1 and 2, results of Wang et al.2021 can be presented with upper-bound of $\alpha(G_t)$ and hence, only require to know this upper-bound instead of the real independence number. This mitigate the "critics" in page 4.
>
> A: You are correct, but the original content in their work write in the form of $\alpha(G_t)$.
>
>
>
> Q: Do the authors choose to run experiments only in comparison with FALCON and not the one of Zhang et al. 2023 because the latter is instable?
>
> A: We appreciate the Zhang's work[1], which firstly deals with the general function space,
> but their method need online regression oracle and works in semi-adversarial setting.
> However, our method only requires offline regression oracle and works in stochastic setting.
> For one thing, the adversarial setting is more difficult than the stochastic setting.
> It usually happens that the algorithms for adversarial settings will have worse regrets than those in the stochastic setting.
> For another, the online oracle is more difficult to design than the offline oracle.
> Online oracle is sensitive to the order of input data sequence while offline oracle not.
> Even for a fixed dataset, the input sequence will also affect the output of online oracles.
> As an example, the OLS estimator is a valid offline oracle but not a valid online oracle.
>
> To see our efforts, we refer the reviewers to option 1.
> Roughly speaking, the option 1 can be viewed as the realization of this method,
> though not fully identical.
> We try to find effective baselines and compare them in theory and simulations.
>
>
> Q: The presented framework uses undirected feedback graph, can this be extended with directed ones? (note that Zhang et al. consider directed graph).
>
> A: This is an interesting work to extend our method to general graphs.
> However, as we admit in the conclusion section in Appendix D,
> we haven't find a way to deal with general graphs.
>
>
> Q: Can we have a definition of the set $S_t$?
>
> A: The name of $S_t$ is called the exploration set.
>
>
> [1] Practical contextual bandits with feedback graphs

---

### Official Review · Reviewer_nWZ2 · 2023-11-04

**Soundness:** 3 good
**Presentation:** 4 excellent
**Contribution:** 3 good
**Rating:** 6
**Confidence:** 3

**Summary:**

The paper tackles the challenging problem of multi-armed bandits with arm rewards in general function spaces and time-varying graph feedback.  The central challenge is the quantification of the graph feedback influence on the regret. To make things harder, the graph changes with time and the reward function doesn't have a closed-form structure that can be exploited.

The authors propose the algorithm FALCON.G to tackle the MAB with a probabilistic time-varying graph feedback problem.  The authors provide both gap-dependent and gap-independent regret bounds and provide matching lower bounds to showcase the optimality of the same. This is supplemented by simulation evidence showcasing the prowess of the proposed methods

**Strengths:**

The paper exhibits several commendable strengths.
1. The authors have done an exceptional job in comparing themselves with other closely related works, ensuring that they improve on the subject matter with this paper. This meticulous attention to detail provides readers with a comprehensive understanding of the existing literature in the field.
2. The proposed method FALCON.G is theoretically proven to be optimal by showing matching upper and lower bounds. Showcasing the dependence on $\delta_f(\cdot)$ for both bounds adds strength to the tightness argument.
3. Both routine of "Option 1" and "Option 2" setups not only demonstrate practicality but also significantly enhances the complexity of problem-solving as compared to previous works.
4. Overall the clarity and coherence of the writing make the paper accessible and easy to follow.

**Weaknesses:**

I would really appreciate the author's comments on the following:

1. **UCB-like approach**: The necessity for forced exploration is not clearly justified. The possibility of employing a UCB (Upper Confidence Bound) type scheme is not explored in depth. It would be beneficial if the authors could provide an explanation of the challenges or limitations associated with the UCB approach.
2. **Offline Oracle**: The usage of regression oracle in FALCON.G resembles (in my opinion) batch regression rather than offline regression.  Especially when considering that FALCON.G utilizes it in the inner loop, albeit not at every cycle. Would you strengthen your argument on the usage of the "offline regression"? Also, what would be the typical complexity for solving the regression problem or would it be an artifact of the functional form of rewards?
3. **Fundamental importance of $\delta_f(\cdot)$**: Would really appreciate a section on the discussion as to whether this graph parameter $\delta_f(\cdot)$ is fundamental to the problem or is it just an artifact of the design of FALCON.G and proof methodology.
4. **Real-world dataset**: Would you see any impending issues for running the simulations on a real-world dataset or dataset with much higher dimensions?

I am willing to change the score after responses from the authors on the above concerns.

**Questions:**

The paper, while detailed, leaves a few questions unanswered:

1. Theorem 3.1's phrasing presents a contradiction. On one hand, it mentions that the "expectation of regret is taken w.r.t. all kinds of randomness," but then goes on to state the result is "with high probability." What is the specific randomness associated with the high probability argument, and how does it differ from the randomness tied to the expectation? Could you clarify this split?

2. It would be beneficial to have real-world examples that align with the setup described in the paper. Specifically, are there tangible instances where the function class and changing graphs over time can be observed?

3. Regarding Theorem 3.2, how expansive is the function class? Are there any practical applications or examples that fall within these classes that can provide a clearer context?

4. In the simulations section, is it feasible to use the baseline of algorithms from related works for comparison? This would offer a more holistic view of how the proposed methods stack up against existing solutions.

---

> ### Author Response · Authors · 2023-11-22
>
> Q: the forced exploration and the construction of the exploration set $S_t$
>
> A: The forced exploration is necessary to avoid suboptimal greedy policy.
> We need to control the size of the exploration set to reduce regret order.
>
>
>
>
> Q: UCB-like approach
>
> A: It is widely known that UCB can achieve comparable performance with sampling method.
> However, it is unclear how one can build valid confidence upper bounds on general function space.
> It is also unknown whether one can truncate the action set as we do.
> To point out the challenges, the previous works [1,2,3] about general function space all focus on IGW rather than UCB.
>
>
> [1] Practical contextual bandit with regression oracle
> [2] Bypassing the monster: A faster and simpler optimal algorithm for contextual bandits under realizability
> [3] Instance-dependent complexity of contextual bandits and reinforcement learning: A disagreement-based perspective
>
>
>
> Q: Fundamental importance of $\delta_f(G)$
>
> A: This is a crucial factor in stochastic graph feedback setting, which dates back to [1].
> It control the minimal size of arms which should be explored.
> Such factor can be obtained by solving LP
> \begin{equation}
>     \begin{aligned}
>     & \min \mathbf{1}^\top \mathbf{z} \\
>     \text{s.t.} \ \ \ & G_t \mathbf{z} \geq \mathbf{1} \\
>     & \mathbf{z} \geq \mathbf{0}.
>     \end{aligned}
> \end{equation}
> The LP can be interpreted as following.
> If one want to explore all arms at least once, how many arms should one pull given the graph $G$.
> The LP can help control the number of exploration in a most efficient way.
> The appearance of this factor in upper and lower bounds[1,2] also show its importance in analysis.
>
>
>
> [1] Reward maximization under uncertainty: leveraging side-observations on networks
> [2] Contextual bandit with side-observations
>
>
>
>
>
> Q: It would be beneficial to have real-world examples that align with the setup described in the paper. Specifically, are there tangible instances where the function class and changing graphs over time can be observed?
>
> A: We haven't found any open dataset which fully fit our setting.
> Though we can conduct experiments in real-world datasets, what we obtain is a regret curve.
> To illustrate the basic property, we believe that the randomly generalized contexts and graphs are enough.
>
>
>
> Q: Theorem 3.1's phrasing presents a contradiction. On one hand, it mentions that the "expectation of regret is taken w.r.t. all kinds of randomness," but then goes on to state the result is "with high probability." What is the specific randomness associated with the high probability argument, and how does it differ from the randomness tied to the expectation? Could you clarify this split?
>
> A: Thanks to point out our typos. The expectation is taken with respect to the randomness of algorithms.
> If one wants to get the expected regret bounds, one can simply set $\delta=1/T$ and it only leads to additional $\delta T$ regrets, which is a constant.
>
>
> Q: Regarding Theorem 3.2, how expansive is the function class? Are there any practical applications or examples that fall within these classes that can provide a clearer context?
>
> A: Theorem 3.2 provides a lower bound. Typically, such bound is proved by construction to show the difficulty level of this problem.
> If one cares about practical applications, the gap-dependent upper bound in Theorem 3.3 might be more meaningful,
> as it holds for general gap $\Delta$ and general function space $\mathcal{F}$.
>
>
> Q: In the simulations section, is it feasible to use the baseline of algorithms from related works for comparison? This would offer a more holistic view of how the proposed methods stack up against existing solutions.
>
> A: For our mentioned works, their settings are not fully identical to us.
> [1] works in MAB with fixed graphs.
> [2] works for linear contextual graphs with fixed graphs.
> [3] deals with the general function space, but their method need online regression oracle and works in semi-adversarial setting.
> However, our method only requires offline regression oracle and works in stochastic setting.
> Roughly speaking, the option 1 can be viewed as the realization of this method,
> though not fully identical.
> To compare them effectively, we have to illustrate the theoretical properties of this work in Table 1.
>
>
>
>
> [1] Reward maximization under uncertainty: leveraging side-observations on networks
> [2] Contextual bandit with side-observations
> [3] Practical contextual bandits with feedback graphs

---

### Meta-Review · Area_Chair_N2ML · 2023-12-03

**Metareview:**

**Summary:**

The paper introduces a novel algorithm for the stochastic contextual bandit problem with graph feedback, aiming to achieve improved regret bounds. The authors claim that their method, FALCON.G, offers both gap-dependent and gap-independent regret bounds, and it outperforms existing algorithms by adapting to time-varying graph structures without requiring prior knowledge of specific graph parameters. Additionally, the paper asserts the minimax optimality of the algorithm through upper and lower bounds, supplemented by numerical experiments.

**Strengths:**
1. The paper tackles an important and challenging problem within the multi-armed bandit domain.

2. The authors provide a comprehensive comparison with existing literature, enhancing the context and relevance of their work.

**Weaknesses:**

1. One reviewer raises a critical concern about the regret upper bound claimed in the paper, suggesting that it might contradict established lower bounds in the field. Unfortunately, the authors did not provide any response. This potential oversight could fundamentally undermine the paper's theoretical contributions.

2. The paper's structure and presentation are noted to be less effective, with key notions and theorems not being clearly or logically introduced.

3.  The paper presents two options for constructing the exploration set without sufficient justification for the need of both, potentially complicating the algorithm unnecessarily.

**Justification For Why Not Higher Score:**

Given the serious concerns raised by the reviewers, particularly regarding the possible theoretical error pointed out by the first reviewer, it seems that the paper may require significant revisions and clarifications to address these issues. The lack of a response from the authors to this critical point is concerning and suggests that the paper might not be ready for publication in its current form.

**Justification For Why Not Lower Score:**

N/A

---

### Decision · Program_Chairs · 2024-01-16

Reject